# Evapotranspiration seasonality across the Amazon basin

Eduardo Eiji Maeda[1], Xuanlong Ma[2], Fabien Wagner[3], Hyungjun Kim[4], Taikan Oki[4], Derek Eamus[5], Alfredo Huete[2]

[1]University of Helsinki, Fisheries and Environmental Management Group, Department of Environmental Sciences, P.O. Box 68, FI-00014, Helsinki, Finland
[2]Climate Change Cluster (C3), University of Technology Sydney, Broadway, New South Wales, 2007, Australia
[3]National Institute for Space Research (INPE), Avenida dos Astronautas 1758, São Jose dos Campos-SP, Brazil
[4]Institute of Industrial Science, The University of Tokyo, Tokyo, Japan
[5] School of Life Sciences, University of Technology Sydney, Broadway, New South Wales, 2007, Australia

*Correspondence to*: Eduardo E. Maeda (eduardo.maeda@helsinki.fi)

**Abstract.** Evapotranspiration (ET) of Amazon forests is a main driver of regional climate patterns and an important indicator of ecosystem functioning. Despite its importance, the seasonal variability of ET over Amazon forests, and its relationship with environmental drivers, is still poorly understood. In this study, we carry out a water balance approach to analyse seasonal patterns in ET and their relationships with water and energy drivers over five sub-basins across the Amazon basin. We used in-situ measurements of river discharge, and remotely sensed estimates of terrestrial water storage, rainfall, and solar radiation. We show that the characteristics of ET seasonality in all sub-basins differ in timing and magnitude. The highest mean annual ET was found in the northern Rio Negro basin (~1497 mm year$^{-1}$) and the lowest values in the Solimões River basin (~986 mm year$^{-1}$). For the first time in a basin-scale study, using observational data, we show that factors limiting ET vary across climatic gradients in the Amazon, confirming local-scale eddy covariance studies. Both annual mean and seasonality in ET are driven by a combination of energy and water availability, as neither rainfall nor radiation alone could explain patterns in ET. In southern basins, despite seasonal rainfall deficits, deep root water uptake allows increasing rates of ET during the dry season, when radiation is usually higher than in the wet season. We demonstrate contrasting ET seasonality with satellite greenness across Amazon forests, with strong asynchronous relationships in ever-wet watersheds, and positive correlations observed in seasonally dry watersheds. Finally, we compared our results with estimates obtained by two ET models, and we conclude that neither of the two tested models could provide a consistent representation of ET seasonal patterns across the Amazon.

## 1. Introduction

Evapotranspiration (ET) in the Amazon rainforest exerts large influences on regional and global climate patterns (Spracklen et al., 2012). Although exact figures vary, it is broadly known that the Amazon River basin transfers massive volumes of water from the land surface to the atmosphere every day, thereby having massive influence on the global energy budget (Aragão, 2012; Christoffersen et al., 2014; Hasler and Avissar, 2007; Restrepo-Coupe et al., 2016). ET is also an indicator of ecosystem functioning, given its intrinsic association with $CO_2$ fluxes during the transpiration process. Hence, any modification of ET over Amazon tropical forests would likely alter the global carbon cycle and further feedback to the rate of a changing climate. Nonetheless, the spatial and temporal characteristics of ET across the Amazon basin, as well as the relative contribution of the multiple drivers to this process, are still uncertain. This may be attributed to the lack of high quality validation data over the full range of ecoregions across the basin, and the thus far unclear influence of climate on vegetation functioning. Recent studies suggested that vegetation phenology, as indicated by leaf demography (Lopes et al., 2016; Restrepo-Coupe et al., 2013; Wu et al., 2016), further increases the complexity of quantifying the relative importance of biotic and abiotic drivers of ecosystem functioning over the Amazon. These uncertainties are reflected in simulations by land surface models (LSMs) and global circulation models (GCMs), hindering the delineation of more reliable climate change scenarios (Karam and Bras, 2008; Restrepo-Coupe et al., 2013, 2016; Werth and Avissar, 2004).

Comprehensive assessments on ET have recently been carried out at local scales using eddy-covariance (EC) methods, which substantially contributed to the understanding of ET seasonality and its drivers in the Amazon (Christoffersen et al., 2014; Fisher et al., 2009; Hasler and Avissar, 2007). EC assessments are, however, limited to small areas. Due to the diversity of vegetation and climatic conditions across the Amazon basin, EC measurements cannot provide a broader overview of the spatial characteristics of ET across the region. The most comprehensive studies carried out so far are based on the data from five to seven flux towers (Christoffersen et al., 2014; Fisher et al., 2009), which although distributed in different ecoregions, cannot represent the full complexity of the Amazon basin. For instance, none of these towers is located in the western Amazon, or in the very wet Rio Negro basin. Furthermore, some sub-basins are characterized by a complex mosaic of land cover types and ecotones, making it impossible to describe the total ET based on unevenly distributed measurements.

Although hydrometeorological models have been implemented to provide spatially explicit assessments of ET in the Amazon, the poor understanding of drivers of ecosystem functioning hinder a more robust parameterization of models (Han et al., 2010). For instance, the spatio-temporal variation of ET is strongly linked to how vegetation assimilates available energy and water (Hasler and Avissar, 2007; Nepstad et al., 1994), a process which just recently started being elucidated (Restrepo-Coupe et al., 2013; Wu et al., 2016). Hence, generally ET models are shown to perform poorly in Amazon forest ecosystems (Karam and Bras, 2008; Restrepo-Coupe et al., 2016; Werth and Avissar, 2004).

Given these bottlenecks, a better understanding of ET seasonality, as well as its relationship with key climate forcings, are needed before model results can be reliably evaluated across the entire Amazon Basin. Water balance approaches are useful

in these situations, as they do not necessarily rely on model assumptions and calibration, and therefore can be applied when there is a lack of in situ ET data or when the drivers of the ET process are not fully understood.

ET assessments using water balance methods have also been undertaken in the Amazon basin, though generally these studies treated the Amazon basin as a whole (Karam and Bras, 2008; Ramillien et al., 2006; Werth and Avissar, 2004). Given the large scale of previous studies, assessments on the drivers of ET have in some cases been inconclusive (e.g. Werth and Avissar, 2004) or reached a single solution for the entire Amazon basin. For instance, Karam and Bras (2009) concluded that Amazonian ET is primarily limited by energy availability. Studies have also been undertaken at smaller scales in neighbouring river basins. Rodell et al. (2011) applied the water balance approach to estimate ET over the Tocantins River basin and found that the seasonal cycle of ET in that basin is weak. These results provide important advances in our understanding of water and energy balance in the Amazon region, but more refined studies are necessary to resolve regional variations. Consequently, water balance assessments at smaller sub-basin scales are needed to evaluate ET limiting factors and their seasonality over a larger range of bioclimatic condition.

Given that plant transpiration is associated with $CO_2$ absorption through leaf stomata, ET is closely linked to ecosystem gross primary production (GPP). For this reason, remotely sensed proxies of photosynthetic activity, in particular vegetation indices (VIs), have often been incorporated into models of ET (e.g. Glenn et al., 2010; Yang et al., 2013). Assessing the relationships between ET and vegetation greenness measured by VIs can also lead to a better understanding of vegetation phenology determinants of ET and ecosystem functioning in general, fostering the improvement of model parameterization. However, studies have found contrasting results on the relationship between canopy greenness measured by VIs and GPP patterns in Amazon forests (Huete et al., 2006; Jones et al., 2014; Maeda et al., 2014). Recent assessments helped clarify this discrepancy, showing that in some parts of the Amazon GPP is driven by the synchronization of new leaf growth with dry season litterfall, increasing the proportion of younger and more light-use efficient leaves, highlighting the importance of leaf phenology (Wu et al., 2016).

The objective of this study was to utilize a water-balance approach to describe seasonal patterns of watershed scale ET across Amazon forests, and relate seasonal patterns with climatic drivers and vegetation greenness. The research questions addressed were: (1) How do seasonal patterns of ET vary across five sub-basins of the Amazon basin? (2) Are the environmental controls of ET similar among sub-basins and across time? (3) How does ET seasonality relate with greenness seasonality? Finally, we compare our ET results with those estimated by a LSM and remote sensing based ET retrievals.

## 2. Material and methods

### 2.1. Evapotranspiration calculation using water-balance approach

The assessments were carried out at watershed level, considering the drainage area of the five major rivers inside the Amazon basin: the Negro, Solimões, Purus, Madeira and Tapajós Rivers (Figure 1). These basins are distributed within different

ecoregions inside the Amazon basin. The size and number of sub-basins were, however, limited by the availability of reliable river discharge data, which is a critical element for the water balance calculation. The ET in each watershed was calculated using the following water budget equation:

$$ET = P - R - \frac{dS}{dt} \qquad (1)$$

where $ET$ is the monthly evapotranspiration, $P$ is the monthly rainfall, $R$ is the river discharge and $dS/dt$ is the change in terrestrial water storage. All units are in mm month$^{-1}$.

Changes in water storage ($dS$) were calculated using Total Water Storage Anomalies (TWSA) estimated from NASA's Gravity Recovery and Climate Experiment (GRACE) satellites (Landerer and Swenson, 2012; Rodell et al., 2004a, 2011; Tapley et al., 2004) using the following equation:

$$dS_n = (TWSA_{n+1} - TWSA_{n-1}) \qquad (2)$$

where $TWSA_{n-1}$ and $TWSA_{n+1}$ are the TWSA values, in mm, for the months preceding and succeeding month $n$, respectively. Hence, the $dS$ computation followed a centered difference approach, which contributes to reduce high-frequency artifacts in the GRACE data (Landerer et al., 2010). To account for the inherent temporal sampling of GRACE, $dS$ values were divided by $dt$, which was calculated by counting the number of days between GRACE observations, and then multiplying by the

number of days in month $n$, reaching the final unit in mm month$^{-1}$.

To facilitate the visualization of ET seasonal patterns, ET for each month was calculated using a three-month sliding window. Hence, the changes in water storage for a certain month were assessed by evaluating the changes in TWSA between the previous and following month (equation 2). For this, linear interpolation was used to adjust the monthly average GRACE TWSA values for the beginning of month n-1, and end of month n+1, resulting in a $dt$ of 3 months, consistent with the three-

month sliding window. The rainfall and river discharge were then calculated accordingly, providing the accumulated volumes inside the three-month window period.

Three monthly GRACE solutions, from different processing centers, were used to compile monthly TWSA: the GFZ (GeoforschungsZentrum Potsdam), CSR (Center for Space Research at University of Texas, Austin), and JPL (Jet Propulsion Laboratory) (Landerer and Swenson, 2012). The three solutions were combined by simple arithmetic mean of the gravity

fields, which according to recent studies is the most effective approach for reducing the noise in the gravity field solutions (Sakumura et al., 2014). The GRACE data were corrected for attenuations on surface mass variations at small spatial scales by multiplying the solution grids by a scaling factor grid provided with the dataset (Landerer and Swenson, 2012).

Rainfall data were obtained from the TRMM 3B43 V7 product. The 3B43 V7 product consists of monthly average precipitation rate (mm hr$^{-1}$), at 0.25° x 0.25° spatial resolution, which combines the estimates generated by sensors on board of the TRMM,

geostationary satellites and ground data (Huffman et al., 2007). The ground data were obtained from NOAA's Climate Anomaly Monitoring System (CAMS), and the global rain gauge product produced by the Global Precipitation Climatology Center (GPCC) (Huffman et al., 2007). Monthly river discharge measurements were obtained from the Environmental

Research Observatory (ORE) HYBAM (Geodynamical, hydrological and biogeochemical control of erosion/alteration and material transport in the Amazon basin).

Uncertainty in the monthly estimates of ET was determined by combining measurement errors on P, R, and dS/dt. Assuming that these variables are independent and normally distributed, the ET relative uncertainty is quantified following the approach proposed by Rodell et al. (2004a, 2011):

$$\upsilon_{ET} = \frac{\sqrt{\upsilon_P^2 P^2 + \upsilon_R^2 R^2 + \upsilon_{dS}^2 dS/dt^2}}{P - R - dS/dt} \tag{3}$$

where $\upsilon$ is the relative uncertainty for each component. The 95% confidence limits were then computed as $\pm\upsilon_{ET}ET$ (Rodell et al., 2011).

Errors in GRACE TWSA estimates were assessed using gridded fields of measurement and leakage errors provided with GRC Tellus data (Landerer and Swenson, 2012). Measurement errors are those related to instrument and signal retrieval errors, while leakage errors are associated with the low spatial resolution of GRACE, as well as spatial smoothing procedures (Rodell et al., 2011). Since errors in nearby pixels are correlated, the calculation of the total error in a region of adjacent pixels needs to account for covariance. Hence, the monthly TWSA errors for each basin were estimated using an algorithm for calculating correlated errors described in https://grace.jpl.nasa.gov/data/get-data/monthly-mass-grids-land/. These values were then multiplied by $\sqrt{2}$ to determine the absolute error of dS/dt, therefore accounting for errors from each of the two consecutive monthly TWSA used for calculating dS (Rodell et al., 2011). Uncertainties in monthly rainfall values ($\upsilon_P$) were assessed using the rainfall relative error layer available in the TRMM 3B43 product (Huffman, 1997). A relative uncertainty of 5% was used for river discharge volumes, as suggested in Rodell et al. (2004a).

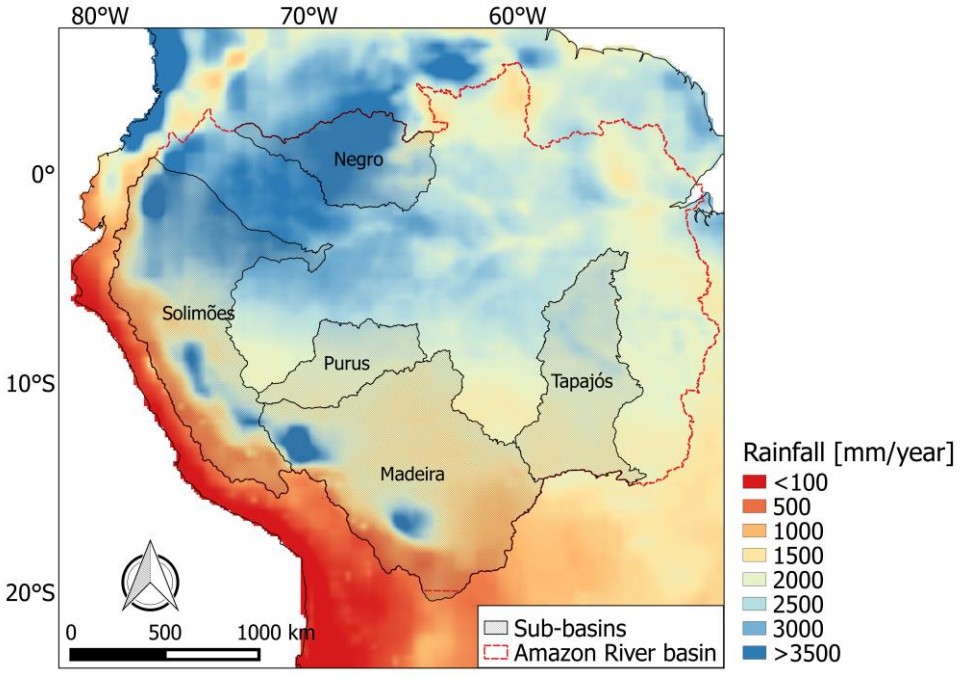

**Figure 1. Amazon River sub-basins assessed in this study. The background map shows the mean annual rainfall 2001-2014, measured by the Tropical Rainfall Measuring Mission (TRMM). The extents of five sub-basins analyzed here are indicated on the map with solid black lines and shading. The solid red line indicates the boundary of the entire Amazon River basin.**

## 2.2. Climate drivers of ET

We evaluate the influence of energy and water input on ET seasonal patterns across all sub-basins. Monthly incident shortwave radiation flux data were obtained from CERES SYN1deg product, version 3A (Kato et al., 2011). Shortwave radiation refers to radiant energy with wavelengths in the visible, near-ultraviolet, and near-infrared spectra. The SYN1deg product provides

10    radiation variables calculated for all-sky, clear-sky, pristine (clear-sky without aerosols), and all-sky without aerosol conditions. In this study, we used the product made for all-sky. The incident radiation flux from SYN1deg product was shown to have a good relationship with photosynthetically active radiation (PAR) measured at flux towers in central Amazon (Maeda et al., 2014). For a better physical interpretation of the results the radiation unit was converted from W m$^{-2}$ to equivalent evaporation in mm month$^{-1}$ by applying a conversion factor equal to the inverse of the latent heat of vaporization (Allen et al.,

15    1998). Monthly rainfall values were obtained from the TRMM 3B43 product, as described in the previous section.

The influence of climate forcings on ET seasonal patterns was assessed using a modified Budyko analysis (Chen et al., 2013; Du et al., 2016). The original Budyko framework (Budyko, 1958) was created to describe the links between climate and catchment hydrological components, resulting in what is known as the "Budyko curve". In this framework, ET is limited by the supply of either water or energy. The type and degree of limitation is determined by the dryness index, which is the ratio

of potential ET (PET) to rainfall (P). The PET provides a proxy of the available energy, and represents the maximum possible value of evapotranspiration under given conditions. Hence, dryness indices lower than 1 represent energy-limited environments, while values higher than 1, water-limited (Budyko, 1958; Donohue et al., 2007). Monthly PET estimates were obtained from the MODIS MOD16A2 (collection 5) product (Mu et al., 2007). In MOD16 product, PET is calculated using

the Penman-Monteith equation driven by surface and remote sensing derived input (Cleugh et al., 2007; Mu et al., 2007).

The other component of the Budyko framework is the evaporative index (ET/P), which describes the partitioning of P into ET and R. In this case, R is proportional to the distance between the curve and a water limit line (i.e. evaporative index=1) and sensible heat is proportional to the distance between the curve and an energy limit line (i.e. when evaporative index=dryness index) (Budyko, 1958; Donohue et al., 2007).

However, these approximations can only be used at steady-state conditions, assuming dS~0. Hence, the original Budyko framework is usually recommended for annual or longer time-scales. For shorter time-scales, studies have shown that intra-annual water storage change should be considered to properly represent the ratio between ET and R (Wang et al., 2009; Zhang et al., 2008). The difference between rainfall and storage change was shown to be a good approach for representing effective precipitation in seasonal models (Chen et al., 2013; Du et al., 2016). Here, we follow this modified Budyko framework, in

which the effective precipitation is represented by P-dS, so that the evaporative index is ET/(P-dS) and the dryness index is PET/(P-dS).

## 2.3. Vegetation greenness proxy

Seasonal patterns of vegetation greenness were assessed using the enhanced vegetation index (EVI) obtained from the

Moderate Resolution Imaging Spectroradiometer (MODIS) (Huete et al., 2002). For this study we used the MODIS MAIAC product, which is processed using MODIS Collection 6 Level 1B (calibrated and geometrically corrected) observations. MAIAC uses an adaptive time series analysis and processing of groups of pixels for advanced cloud detection, aerosol retrievals and atmospheric correction (Lyapustin et al., 2012). This dataset provides geometrically-normalized spectral reflectances (BRFn), which were used in this study. EVI was calculated considering a fixed sun-sensor geometry, with sun

zenith angle of 45 degrees and nadir view angle. We used observations from the Terra and Aqua satellites collected between 2001 and 2012, and data were obtained from the Atmosphere Archive and Distribution System (LAADS Web: ftp://ladsweb.nascom.nasa.gov/MAIAC).

## 2.4. Comparison with modelled ET

We compare our ET estimates with two model-based estimates. The first modelled ET dataset was obtained from the NOAH 2.7.1 Land Surface Model (LSM) in the Global Land Data Assimilation System (GLDAS) (Rodell et al., 2004b). The data have a 0.25 ° spatial resolution and the temporal resolution is monthly. The NOAH LSM comprises three components of latent

heat: bare soil evaporation, transpiration, and evaporation from canopy-intercepted water (Chen et al. 1996, Ek et al. 2003). Bare soil evaporation and wet canopy evaporation are calculated scaling potential evaporation by soil moisture saturation in the upper soil layer and saturation of canopy interception, respectively. Transpiration is determined by potential evaporation, canopy resistance including soil moisture stress, and canopy wetness. Potential evaporation is calculated by Penman approach of Mahrt and Ek (1984)

The second modeled ET dataset was obtained from the MODIS MOD16A2 product (Mu et al., 2007). The MOD16 ET is calculated by a modified Penman–Monteith ET method, which uses ground-based meteorological observations and remote sensing data from MODIS to provide global estimates of ET. For both modeled ET datasets, NOAH and MOD16, data were obtained from January 2001 to December 2014.

## 3. Results

### 3.1. Spatial and seasonal variations in ET across five Amazon sub-basins

A summary of the components used for the water balance equation (eq 1), for the period between 2001 and 2014, are presented in Table 1. The largest river discharge and rainfall volumes were observed in the Rio Negro basin, with an annual mean of 1692 mm year$^{-1}$ and 3285 mm year$^{-1}$, respectively. The lowest values were observed in the Madeira River, where mean discharge was 584 mm year$^{-1}$ and mean rainfall 1716 mm year$^{-1}$ (Table 1). Seasonal variations in total water storage are larger in the Tapajós River basin, where the mean maximum was 132 mm month$^{-1}$ (i.e. increasing water storage) and mean minimum was -123 mm month$^{-1}$ (i.e. decreasing water storage) (Table 1).

**Table 1. Summary of the river discharge, rainfall and dS/dT in the five sub-basins analyzed in this study. For each variable, the monthly average maximum and minimum, as well as the annual mean, are presented. All values are averages for the period between 2001 and 2014. Long-term annual averages of dS/dT are generally close to zero, and therefore not presented.**

| | Mean values (2001-2014) | Negro | Solimões | Purus | Madeira | Tapajós |
|---|---|---|---|---|---|---|
| Discharge (R) | Monthly Max [mm month$^{-1}$] | 213 | 138 | 123 | 84 | 117 |
| | Monthly Min [mm month$^{-1}$] | 96 | 63 | 15 | 12 | 24 |
| | **Mean annual [mm year$^{-1}$]** | **1692** | **1241** | **767** | **584** | **767** |
| Rainfall (P) | Monthly Max [mm month$^{-1}$] | 360 | 234 | 294 | 252 | 327 |
| | Monthly Min [mm month$^{-1}$] | 213 | 123 | 45 | 39 | 21 |
| | **Mean annual [mm year$^{-1}$]** | **3285** | **2227** | **2154** | **1716** | **2154** |
| dS/dT | Monthly Max [mm month$^{-1}$] | 48 | 54 | 99 | 87 | 132 |
| | Monthly Min [mm month$^{-1}$] | -45 | -72 | -96 | -75 | -123 |
| ET | Monthly Max [mm month$^{-1}$] | 132 | 105 | 138 | 114 | 123 |
| | Monthly Min [mm month$^{-1}$] | 108 | 63 | 90 | 78 | 99 |
| | **Mean annual [mm year$^{-1}$]** | **1497** | **986** | **1351** | **1132** | **1314** |

Annual mean ET values varied among five sub-basins (Table 1; Figure 2). The largest mean annual ET was observed in the Rio Negro basin (~1497 mm year$^{-1}$), while the lowest value was observed in the Solimões River basin (~986 mm year$^{-1}$) (Table 1; Figure 2). The relative magnitude of mean ET among the Negro, Purus, Madeira and Tapajós basins are consistent with rainfall variation within these regions, i.e., the highest mean annual ET corresponds to the highest mean annual rainfall, and vice versa (Figure 2). The Solimões basin, however, is an exception. Despite having annual average rainfall similar to what was observed in Purus, its mean ET rates were significantly smaller (Figure 2). This may be explained by the lower average solar radiation inside the Solimões basin, with an annual average of 2480 mm year$^{-1}$, while the average in the Purus basin was 2570 mm year$^{-1}$ (Figure 2). Furthermore, portions of the Solimões basin are located in the Andes region, which is characterized by higher altitudes, lower rainfall and sparse vegetation (Figure 1).

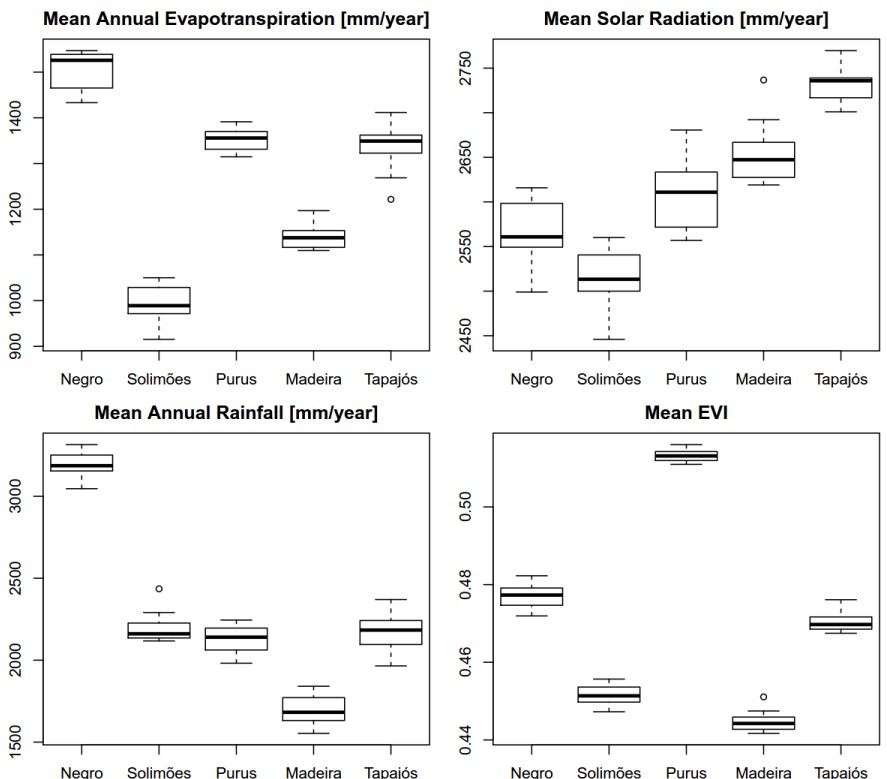

**Figure 2. Boxplots with mean annual evapotranspiration, solar radiation, rainfall and EVI for the five sub-basins analysed in the study for the period 2001 – 2014 inclusive.**

The seasonal patterns of rainfall, radiation and ET are presented in Figure 3. Seasonal variation of ET is observed in Solimões, Purus, Madeira and Tapajós, but less evident in the Rio Negro basin. In the Solimões basin, ET was highest in September and October, while the lowest values were observed in December and January (Figure 3). In the Purus, Madeira and Tapajós basins, ET peaks around November, February and November, respectively (Figure 3). The uncertainty on ET estimates were generally higher during the rainy seasons. i.e. approximately March-July at the Negro basin, and November-Abril in all the other sub-basins.

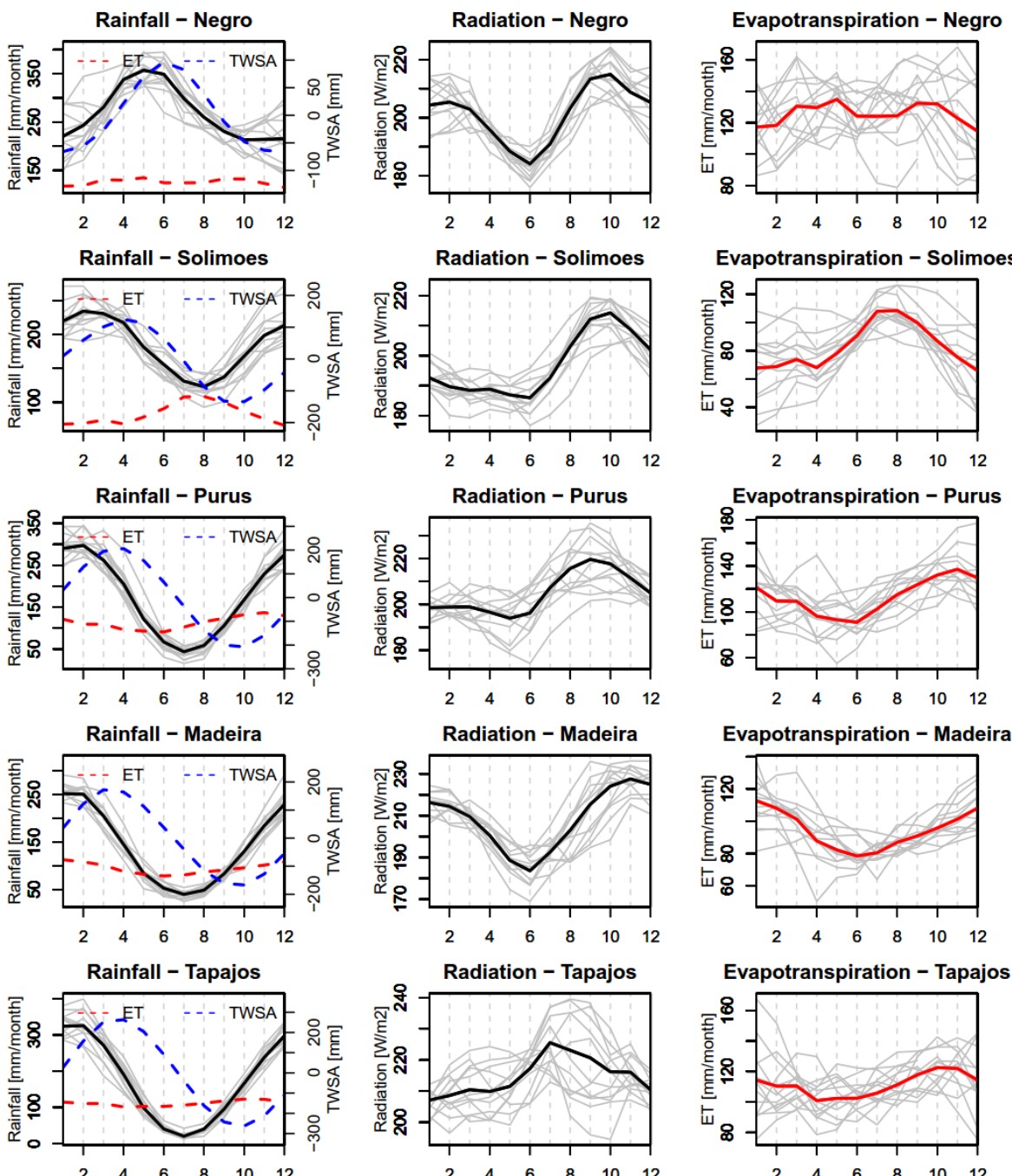

**Figure 3. Seasonal variations of rainfall, radiation and evapotranspiration inside each sub-basin. Gray lines represent the values for each year from 2002-2014, and solid dark lines represent the average values for each month. Months are represented from 1 (January) to 12 (December). The dashed blue line in the first column shows the mean seasonal variation of GRACE terrestrial water storage anomalies (TWSA), and the dashed red line is the mean seasonal variation of water-balance ET, for each sub-basin. Vertical bars indicate the uncertainty in the water budget estimates.**

In terms of long-term average values, ET did not exceed rainfall in any season of the year, in the Negro and Solimões basin sites. This indicates that, under average conditions, ET is not limited by water availability, even in the driest season. In the Purus, Madeira and Tapajós sites, rainfall deficit (i.e. ET>rainfall) was observed between June and August. Water availability, therefore, may be a limiting factor for ET during the dry season, although soil water storage and root access to deep water can potentially compensate the rainfall deficit. In these three basins, the smallest rate of ET was observed in May-June, period in which rainfall volumes are in steady decline. The seasonal patterns of each component used for the water balance calculation, as well as their uncertainties, are presented in Figure S1.

## 3.2. Climatic drivers of Amazon ET seasonality

The modified Budyko analysis of monthly ET values is presented in Figure 4. The dryness index in the Negro basin was consistently below the water limit threshold (<1). For this sub-basin, the water balance analyses show the basin to consistently follow the energy limited line (red dashed line), indicating some degree of energy limitation. However, our results show small seasonal variation of ET in the Negro basin, despite clear intra-annual variation in solar radiation (mean annual amplitude of 30 $W.m^{-2}$) and rainfall (mean annual amplitude of 140 $mm.month^{-1}$). These contrasting results are likely explained by the very high ET rates at the Negro basin (Table 1), which could represent an upper limit in forest water use capacity.

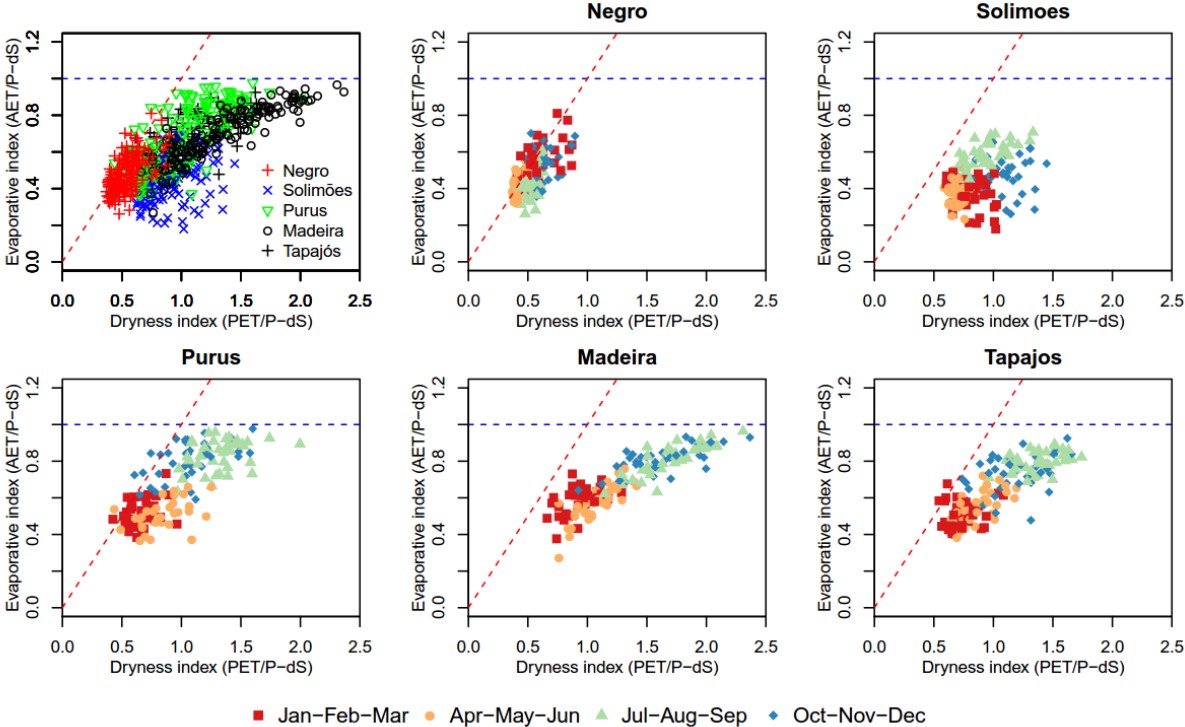

**Figure 4. Modified Budyko analysis for monthly water balance values. The red dashed line represents the energy limitation threshold, above which ET is limited by solar radiation. The blue dashed line represents the water limitation threshold.**

5  In the three southern basins, Purus, Madeira and Tapajós, water limitation was consistently observed during July, August and September (Figure 4). This is consistent with the observation of seasonal rainfall deficits in these regions, but it contrasts with the ET seasonal patterns in these basins (Figure 3). In all southern basins, ET reached the lowest values before the period of minimum rainfall. These results suggest that in the southern Amazon ecotone, deep root water intake plays a key role in maintaining ecosystem productivity during the dry season. In the Purus and Tapajós basins, the Budyko curves are particularly

10  close to the energy limit threshold during January, February and March. This shows that ET in these regions can experience some degree of energy limitation during the wet season. The Solimões basin is shown to be located in a transition region, where water limitation can occur in drier years. The energy constraint in the Solimões basin was also lower than that observed in the Negro basin.  The mean seasonal patterns of PET, used for carrying out the Budyko analysis, are presented in Figure S2. Figure 5 shows a scatterplot of monthly radiation *versus* rainfall, with data points labeled by their corresponding monthly

15  average ET values. This figure reveals a general pattern on the relationships among monthly rainfall, radiation and ET. As expected, lower monthly ET values are consistently observed when both radiation and rainfall are low. However, the lowest ET values are located in the mid-range of both radiation and rainfall. This pattern may reflect the influence of other variables driving ET rates, in particular soil water storage and root access to deep water. For instance, at a radiation range of 200-250

mm month$^{-1}$, ET is minimum (i.e. ET< 80 mm month$^{-1}$) when rainfall is around 200 mm month$^{-1}$, and slightly higher (i.e. ET~100 mm month$^{-1}$) when rainfall drops below 100 mm month$^{-1}$. Hence, these observation are likely from regions where plants have better access to deep water, and can maintain higher ET rates despite reduced rainfall. Interestingly, the highest ET values are not observed when radiation was highest, providing more evidence that in some regions water availability may also be a limiting factor of ET, in combination with radiation.

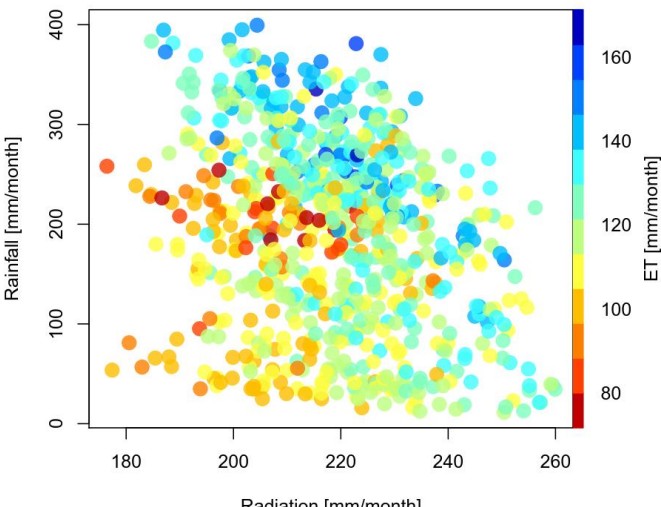

**Figure 5. Scatterplot of monthly radiation and rainfall for the five sub-basins. Colour gradient indicates the monthly ET value, from high (blue) to low (red).**

### 3.3. Relationship between ET and canopy greenness

The relationship between ET and vegetation greenness varied across the Amazon basin (Figure 6 and Table 2). In the Negro basin, no significant relationship was found between EVI and ET. In this region, vegetation greening was observed between September and December, followed by a steady decline in EVI until the following August (Figure 7).

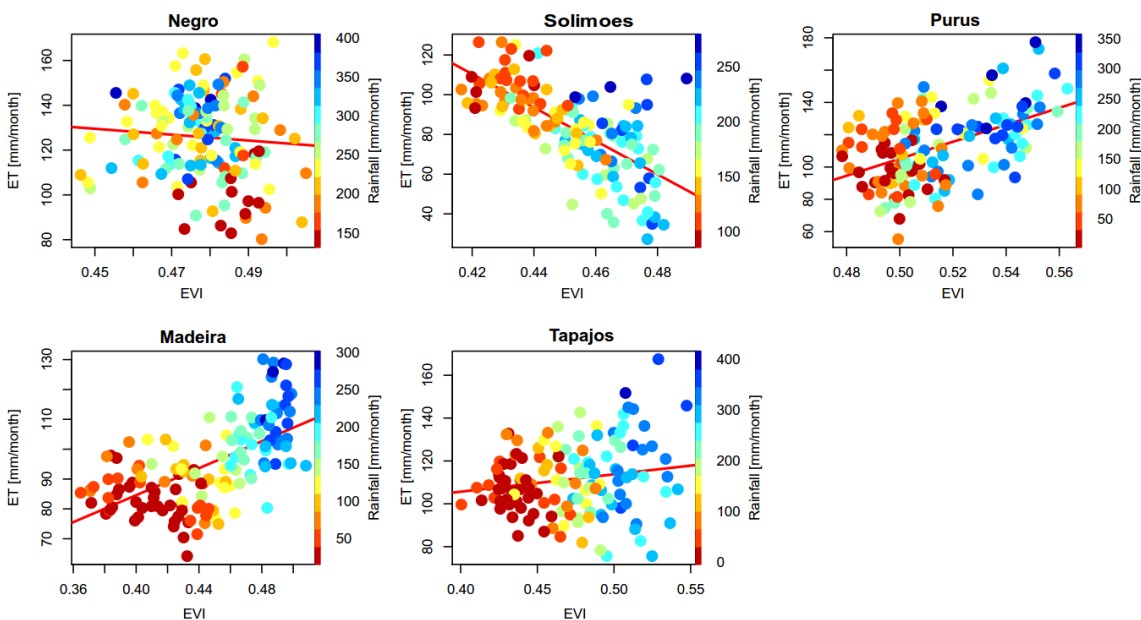

**Figure 6. Relationship between monthly evapotranspiration (ET) and MODIS enhanced vegetation index (EVI) at each Amazon sub-basin using the data from 2001 to 2014. Colour gradient indicates the monthly rainfall value, from high (blue) to low (red).**

Significant positive correlations ($p < 0.05$) between EVI and ET were observed in the Purus, Madeira and Tapajós basins (Figure 6 and Table 2). In these regions, a clear pattern was observed, in which higher ET takes place when vegetation is greener and when rainfall is higher. In the Solimões basin, despite higher EVI values observed during the wet season (Figure 6), an opposite pattern between ET and EVI was observed, i.e. higher ET takes place when EVI is lower. In Solimões, vegetation greening also occurs between September and December, with declining from January until August (Figure 7).

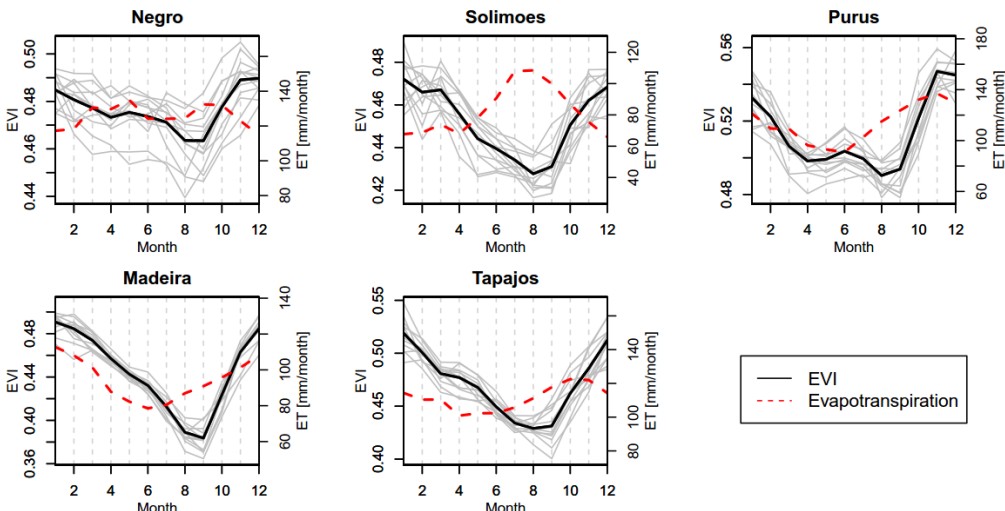

**Figure 7. Seasonal patterns of MODIS EVI in the five Amazon sub-basins. The black lines show the monthly average values from 2001 to 2014, while gray lines show individual monthly values for each year. The mean seasonal variations in ET for each sub-basin are represented as red dashed lines.**

**Table 2. Coefficients of the linear regression between evapotranspiration (ET) and MODIS enhanced vegetation index (EVI) for each of the five sub-basins (* $p<0.05$).**

|  | Intercept | Slope | $R^2$ |
|---|---|---|---|
| Negro | 6.0 | -4.06 | 0.006 |
| Solimões | 14.9 | -27.0 | 0.463* |
| Purus | -5.3 | 17.5 | 0.259* |
| Madeira | -0.4 | 7.9 | 0.383* |
| Tapajós | 2.2 | 3.1 | 0.035* |

### 3.4. Comparison with ET estimated by models

10   We further assessed the ability of two ET models, NOAH-LSM and MOD16 P-M, to replicate the seasonality of ET as derived from observation-based water balance calculation. Our results showed that neither of these two models was able to reproduce the timing and magnitude of seasonal ET patterns as calculated from the water-balance approach (Figure 8). In the Negro basin, NOAH-LSM estimates were consistently below the water balance and MOD16 P-M values, with an annual average of 1241 mm year[-1]. Nonetheless, both model estimates were within the 95% confidence intervals of the water-balance

15   calculations. In the Negro basin, both NOAH-LSM and MOD16 P-M show a decreasing ET trend from January to May, followed by an increasing trend (Figure 8). NOAH-LSM ET reached its maximum in September, while MOD16 P-M ET maximum was observed in October (Figure 8).

In the Solimões basin, NOAH-LSM and MOD16 P-M ET showed similar seasonal patterns, but MOD16 P-M ET values were on average 25 mm month$^{-1}$ larger than the NOAH-LSM estimates throughout the year (Figure 8). Nonetheless, both models showed ET seasonal patterns discrepant with the water balance calculation. Both models indicate highest ET in December/January, when the water balance showed the lowest seasonal values (Figure 8). The MOD16 P-M ET extrapolates the water balance uncertainties between November and April, and NOAH-LSM ET between June and August.

The MOD16 P-M ET showed almost no seasonality in the Purus basin, while NOAH-LSM and water balance ET indicate a decrease in ET during May (Figure 8). However, the NOAH-LSM underestimated the ET recovery in the following months, in particular between August and November (Figure 8). The same pattern was observed in the Madeira and Tapajós basins, where both models show significantly lower ET values in August, September and October (Figure 8), below the 95% confidence limits of the water balance estimates.

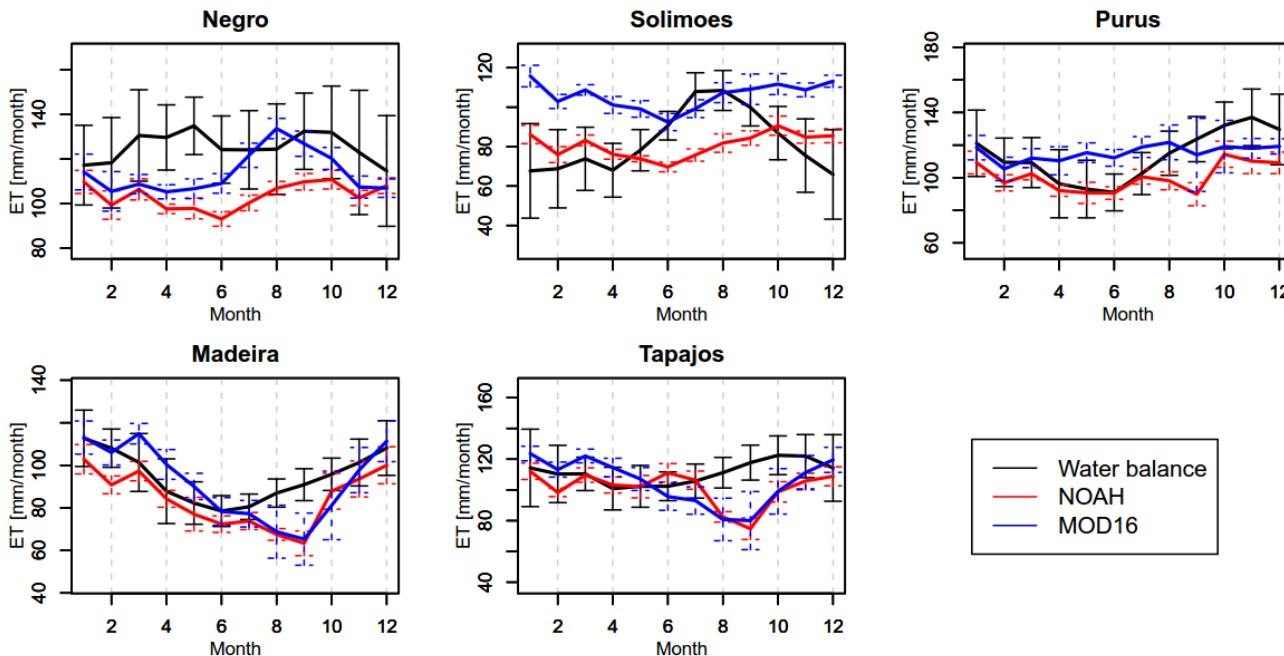

Figure 8. Seasonal ET patterns obtained using the water balance method (black line), NOAH land surface model (red) and MODIS MOD16 P-M model (blue). Vertical bars indicate the ±1 standard deviation of monthly observations from 2001 to 2014. The grey area represents uncertainty in the water budget estimates.

## 4. Discussion

Previous estimates of ET in the Amazon basin vary considerably in terms of magnitude and seasonal patterns. Water balance assessments undertaken at larger scales (e.g. the entire Amazon basin) found mean annual ET estimates varying from 767 mm

year$^{-1}$ to 1642 mm year$^{-1}$ (Callede et al., 2002; Karam and Bras, 2008; Ramillien et al., 2006; Rao et al., 1996; Werth and Avissar, 2004). The ET values we describe for Amazon sub-basins are within this range. We show that in some wet regions, such as the Rio Negro basin, mean annual ET can be above 1400 mm year$^{-1}$, while in southern basins it vary from 1130 mm year$^{-1}$ to 1350 mm year$^{-1}$. Hence, we find that the lower range of 767 mm year$^{-1}$ described in previous studies (Karam and Bras, 2008) is likely to underestimate the average ET for the entire Amazon basin.

Our results show that the seasonal patterns of ET of five sub-basins across the Amazon vary in timing and magnitude. This spatial heterogeneity in ET seasonality is in agreement with previous studies carried out at local scale using EC method (Christoffersen et al., 2014; Fisher et al., 2009). Christoffersen et al. (2014) reported either a flat seasonal cycle or a slight dry season decrease of ET at transitional southern forests, while equatorial forest ET showed ET peaking with net radiation during the dry season. Despite agreeing on the main climatic forcing of ET process across these different ecoregions, our results unveil some differences on the timing of seasonal increases in ET and peak in relation to climatic variables. These differences are discussed in detail bellow.

## 4.1. Climatic drivers of Amazon ET seasonality

Discussions on the drivers of ecosystem function seasonality in the Amazon have often resulted in conflicting results. Our results revealed that in most cases ET seasonality is driven by a balance between radiation, rainfall and vegetation regulations, rather than being exclusively limited by any one of these factors. For instance, the peak timing of ET at five sub-basins did not correspond to the peak timing of either rainfall or radiation, demonstrating that the arbitrary partition of the Amazon basin into either energy-limited or water-limited is unrealistic and would result in large uncertainty in predicted ET patterns, as we showed in this study.

We further demonstrated the degree of radiation and rainfall limitation, as well as their interactive effects on ET based on a modified Budyko analysis (Figures 4-6). Our results show that the evaporative index (ET / (P - dS)) exhibited a positive, nonlinear-type, dependency on climatic dryness index (PET / (P - dS)), which falls well within the modified Budyko framework. The modification of the classic Budyko model is the consideration of temporal changes in water-storage, in which total water-availability for evaporation should be quantified as the sum of monthly precipitation and water-storage change, termed as effective precipitation. Our results thus revealed the importance of considering plant controls in water-balance accounting over Amazon basin forests, as these evergreen trees, with their lengthy root-systems, have the ability to tap deep soil-/ground-water to meet atmospheric water demand.

ET in the Solimões basin presented an earlier peak, in comparison with the other Southern basins, which was out of phase with both radiation and rainfall. Our results indicate that, in Solimões, ET is normally not limited by water or energy input, hence water loss regulation may not be critical for plants. In this case, seasonality of productivity can be regulated to reach an optimization that maximize the use of both available water and energy resources. It is however important considering the relatively high uncertainties in *dS* and rainfall over this region (Figure S1), which can potentially affect seasonal patterns, leading to misinterpretation of the relationships between ET, climate and vegetation.

In the Purus, Tapajós and Madeira basins, which encompass regions often considered to be water limited (Guan et al., 2015; Jones et al., 2014; Xu et al., 2015), ET does not necessarily reach the lowest values during the driest periods (Figure 3). Instead, we found increased ET before the end of the dry season, and ET rates can increase even in rainfall deficit conditions (Figure 4). This pattern can be explained by plants access to deep soil water (Nepstad et al., 1994). This argument is reinforced by the seasonal patterns of TWS demonstrated in Figure 3, which show that in southern basins TWS lags rainfall by approximately three months. Hence, during the meteorological dry season (i.e. when rainfall is low), soil water storage still remains relatively high. When the soils reach their lower storage volumes, 3 months after the peak of dry season, the rainy season has already started, providing water supply to be used by plants.

These results concur with previous findings showing a weak relationship between rainfall anomalies and EVI anomalies (Maeda et al., 2015), indicating a lower sensitivity of ecosystem functioning to rainfall extremes at transition forests in the southern Amazon. Furthermore, we show that besides dealing with seasonal rainfall deficit, southern basins remain limited by radiation energy availability during a certain period of the year (Figure 4), which explains the ET recovery before the driest period, i.e. when radiation starts to increase (Figure 3).

However, it is important to highlight the fact that, although these analyses are based on sub-basins across the Amazon, they still enclose relatively large areas with substantial heterogeneities. In particular, the Madeira and Tapajós basins are characterized by a large latitudinal gradient and, consequently, different ecosystems are present within these sub-basins. Hence, it is likely that, although on average water availability is not critical at the Tapajós and Madeira basins during the dry season, water limitation may occur in southern (drier) parts of these basins.

## 4.2. Relationship between ET and canopy greenness

The biophysical causes of EVI seasonality in Amazon evergreen forests have been intensively discussed in recent years (Bi et al., 2015; Hilker et al., 2015; Maeda et al., 2014; Morton et al., 2014; Myneni et al., 2007). Recent studies indicate that in wet equatorial forests, EVI is driven by a net increase in leaf production (Lopes et al., 2016). The seasonal variation in EVI was shown to be more evident in the dry season, when most plants release old leaves while simultaneously producing new leaves and, therefore, increase EVI.

Furthermore, studies have shown that southern and Equatorial forests have different cues for leaf flushing, i.e. plant growing season is initiated by different climatic factors (Wagner et al., 2016). Hence, our results indicate a decoupling between ET fluxes and seasonal cycles of canopy foliage. In general, relationships were better in southern basins where rainfall deficits were observed, in particular Purus and Madeira. In these cases, the climatic triggers for leaf flushing/litter and productivity drivers are likely to be in phase. In the southern Amazon, leaf growth was shown to be initiated by water input (Wagner et al., 2016), which means that peak greening should be observed some months after the beginning of the wet season. In these regions, ET was found to decline as rainfall decreased between March and May. Nonetheless, ET trends recovered before the peak of

the dry season, increasing with higher solar radiation – suggesting that soil water was available to the trees even during the peak of the dry season.

In the Negro basin, ET was not significantly correlated with EVI, while in the Solimões Basin, ET and EVI were inversely related. In these cases, different mechanisms are likely to drive ET and canopy greenness patterns. In the wet equatorial forests, leaf flushing was shown to be initiated by the increase in solar radiation (Lopes et al., 2016; Wagner et al., 2016). The subsequent decrease in greening, however, follows a different pattern, where a slow decrease in EVI might be associated with leaves aging, epiphylls, herbivores, and leaf fall.

Lags between forest functioning and canopy greening have been previously reported from local scale experiments. Wu et al (2016) suggested that these discrepancies could be explained by leaf demography, given a higher photosynthetic capacity of mature leaves. In other words, while LAI increases during the dry season due to new leaves flushing, young leaves have lower photosynthetic capacity, which gradually increases as leaves become mature – but then declines as leaves senesce (Wu et al., 2016). They, hence conclude that phenology of photosynthetic capacity, and not climate variability, is the main driver of ecosystem productivity (Wu et al., 2016). Our results confirm this decoupling of vegetation functioning and leaf production in wet evergreen forests. Nonetheless, we demonstrate that vegetation function seasonality, as described by sub-basin scale ET, is not independent from climate intra-annual variability.

Our results indicate that, over tropical regions, using EVI as an input variable to ET models should be done with caution. Although EVI and other vegetation indices have been successfully applied for modelling ET in temperate zones (e.g. Yang et al., 2013), we show that the relationship between ET and EVI at wet tropical forests is more complex. Hence, further studies are needed to better understand how ET relates to EVI over a broader latitudinal gradient, and how such variability in the relationship can be incorporated into ET models.

## 4.3. Uncertainties of the water-balance approach and comparison with model estimates

Assessing uncertainties of ET estimates in Amazon forests is challenging, given the lack of reference datasets. Previous studies indicate that ET estimates based on GRACE water balance approach may have higher uncertainties than LSM estimates (Long, 2014). This assessment was, however, carried out in a region with good data quality for model parameterization, and where the drivers of ecosystem functioning are better understood. In the Amazon, where parameterization of models are usually more challenging due to low data quality and unknown biophysical parameters, water balance methods are still considered an adequate alternative.

Our results indicate higher uncertainties for estimating ET based on water-balance approach during the wet seasons. This is primarily caused by the increase in errors on rainfall estimates from TRMM during this period (Figure S1). Although previous studies indicate that uncertainty in $dS$ is typically the dominant component of the error budget (Rodell et al., 2011), we show that, in the Amazon region, rainfall errors are often the main contributor to ET uncertainties, particularly during the wet seasons (Figure 3 and 8). In almost all basins, with the exception of Negro basin, rainfall was the main source of error during the wet

seasons, while during the dry season *dS/dt* was the major source of uncertainty (Figure S1). From the components contributing for TWSA uncertainties, leakage errors were dominant in all basins, while measurement errors were relatively lower.

Assessing ET at local scales, using eddy covariance methods, Christoffersen et al. (2014) concluded that most models are not able to represent ET seasonality at different locations across the Amazon. They argue that models are unable to properly represent canopy dynamics mediated by leaf phenology, which is believed to play a significant role in regulating ET seasonality. Assessing spatially averaged ET for the Amazon basin, Karam and Bras (2008) reported that mean annual values calculated using water balance methods (including Callede et al., 2002; Ramillien et al., 2006) show significantly lower estimates when compared with output from LSMs. Although the models compared in this study are not the same, our results diverge from these claims. At the Negro, Purus, Madeira and Tapajós basins, mean annual ET values calculated with the water balance method were higher than NOAH and MOD16 estimates. Only at the Solimões basin, annual mean ET from MOD16 was higher than the other methods.

ET estimates from NOAH-LSM and MOD16 P-M could not provide a consistent representation of ET seasonality between each other in all sub-basins (Figure 8). Although a full comparison with ET models is beyond the scope of this study, our results confirm that models still disagree with each other in estimating Amazon ET seasonality, indicating uncertainties associated with either input datasets or model assumptions. Both models seem to overestimate water stress in the southern basins, i.e. while models predict a decline in ET after the driest period, the water balance estimate shows an early recovery from the dry season, followed by a steady increase until the end of wet season (Figure 8).

One potential source of uncertainty in the NOAH-LSM estimates is the fractional total vegetation cover ($f_c$), which contributes for defining both transpiration and interception evaporation. In NOAH, $f_c$ seasonal variation is estimated from remotely sensed Normalized Difference Vegetation Index (NDVI) climatology (Gutman and Ignatov, 1998; Marshall et al., 2013). Nonetheless, studies have shown that, due to saturation over dense tropical forests, as well as illumination artefacts, NDVI may not correctly describe seasonal changes in vegetation structure over the Amazon forests (Huete et al., 2002; Maeda et al., 2016).

The PET estimates used for the modified Budyko analysis (Figure 4) is also based on models, and therefore is likely to carry some level of uncertainty. Given that PET is a physical measure of atmospheric water demand, and do not depend on vegetation interactions, the reliability of estimates for the Amazon basin are likely to be the same as for other regions. Having said that, uncertainties in PET and ET have noticeable effects on the derived Budyko curves. For instance, underestimated PET values may lead to dryness index values higher than evaporative index, leading to plotted values that exceed the energy limit line. This is also observed when using an alternative PET dataset (Figure S3). Previous studies, however, reported that monthly-average evaporation may exceed potential estimates by about 10 % during wet months (Shuttleworth, 1988). On the other hand, overestimated PET can lead to misleading conclusions of higher water limitation in Figure 4. This is likely to be the case in the Solimões basin, as the seasonal patterns presented in Figure 3, which are based only on observational data, indicate that in the Solimões basin average rainfall is always higher than average ET. Water limitation conditions in this region are still likely, given inter-annual variability in rainfall and ET, but it should not be a condition that is repeated consistently every year.

## 5. Conclusions

Our results demonstrate strong spatial heterogeneity in ET across five ecoregions within the Amazon basin. Seasonal cycles of ET are shown to vary in timing and magnitude, driven by intra-annual climate variability across sub-basins. Based on a modified Budyko analysis, we show the interactive effects of rainfall, solar radiation and soil water storage on ET fluxes. Nonetheless, our results indicate that neither energy nor water input alone is sufficient to explain ET seasonality across five sub-basins, regardless of the average degree of dryness, demonstrating a dynamic shift in the degree of energy-/water-limitation across space and time. Although eddy covariance studies have shown that ET in the Amazon can be limited by different climatic factors, this fact had not yet been verified at basin scales using observational data.

We demonstrate a decoupling between ET and vegetation greenness seasonal patterns in wet Amazonian forests. While a positive and significant relationships between EVI and ET were observed in southern basins, inverse or not significant correlations were observed in basins located at lower latitudes. This finding indicates that ecosystem models based on remotely sensed vegetation indices, including remote sensing based ET models, need to be further assessed to better represent ecosystem function seasonality in wet tropical forests.

A comparison with two ET models, NOAH-LSM and MOD16 P-M, showed that models are still unable to consistently represent ET seasonal patterns in the Amazon forest. In the Solimões and Negro basins, both models presented a different seasonal pattern when compared with our water balance approach. In southern basins, where rainfall is lower, models seem to overestimate water limitation during the dry season, and therefore underestimate ET.

### Acknowledgments

This study was financially supported by the Academy of Finland (Decision No. 266393). Hyungjun Kim was supported by Japan Society for the Promotion of Science KAKENHI (16H06291). We would like to thank Dr. Alexei I. Lyapustin, from NASA Goddard Space Flight Center, and Dr. Yujie Wang, from the University of Maryland, for their support in processing and distributing the MODIS MAIAC dataset. A. Huete and X. Ma were supported by an Australian Research Council - Discovery Project (ARC-DP140102698, CI Huete). X. Ma was also supported by an Early Career Research Grant (ECRG) from University of Technology Sydney (PRO16-1358, CI Ma).

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
