# Peer review of "Evapotranspiration seasonality across the Amazon basin"

_Earth System Dynamics, 2016_

## Referee Comment (RC1)

**Reviewer's Comments**
Manuscript #
"Evapotranspiration seasonality across the Amazon basin"
by Eduardo Eiji Maeda, Xuanlong Ma, Fabien Wagner, Hyungjun Kim, Taikan Oki,
Derek Eamus, Alfredo Huete

**Summary:**
This study employs a water budget residual approach to estimating evapotranspiration in
five sub-basins of the Amazon River basin, and uses the results to describe the seasonal
cycles of ET in each basin and their relationships with precipitation, solar radiation, and
vegetation indices. Overall this is a well written and interesting paper that will be a
valuable addition to the literature. However, there is a significant error in one of the
equations, as detailed next. Once that error is fixed, the analyses, figures, and discussions
will have to be revised accordingly. While this will require a major revision, I believe it
is fairly straightforward, and once it is completed, along with an error analysis, the paper
should be accepted readily.
Interdisciplinary scope: very good
Scientific merits: very good
Technical quality: fair/good

**Major Comments:**
• Equation 2 is incorrect. It should be
$dS_n = (TWSA_{n+1} - TWSA_{n-1})/((n+1) - (n-1))$, or equivalently,
$dS_n = (TWSA_{n+1} - TWSA_{n-1})/2$,
as shown in equation 13 in Swenson and Wahr (2006). This error leads to dS, and
consequently monthly/seasonal ET, being inaccurately estimated throughout the
manuscript. Consequently, analysis and discussion of the results, including the Budyko
analysis, relationship with vegetation greenness, and comparison with models, must be
revised after the estimates have been fixed.
• There is no uncertainty analysis provided for the water budget ET estimates. It should
be included, and could be accomplished by computing the square root of the sum of the
squares of the P, R, and dS errors. See, for example, Rodell et al. (2011).
Rodell, M., E.B. McWilliams, J.S. Famiglietti, H.K. Beaudoing, and J. Nigro, Estimating
evapotranspiration using an observation based terrestrial water budget, Hydrol. Proc., 25,
4082-4092, doi:10.1002/hyp.8369, 2011.

**Minor Comments:**
• Page 2, line 30 – Add to the literature review that Rodell et al. (2011) applied the water
balance approach (with observed precipitation, runoff, and GRACE terrestrial water
storage) to ET estimation over the Tocantins River basin (among others) and found that
the seasonal cycle of ET in that basin is weak.

**Technical Corrections**
• Page 7, line 6 – "are" should be "is" (the subject is "analysis").

---

## Referee Comment (RC2) · Anonymous Referee #2 · 10 Mar 2017

**Evapotranspiration seasonality across the Amazon basin**

By Eduardo Eiji Maeda, Xuanlong Ma, Fabien Wagner, Hyungjun Kim, Taikan Oki, Derek Eamus, Alfredo Huete

Paper review

**Summary**

This paper uses a water balance approach to calculate evapotranspiration (ET) in five sub-basin of the Amazon River. It focusses on seasonal variability of ET and its relationship with precipitation, radiation and vegetation greenness. It also compares water-balance ET with model-calculated ET. Overall, the paper is very clearly written and is a valuable input to the literature. It confirms the highly variable nature of ET across the Amazon basin, and the difficulty models still have to capture ET seasonality in that region. However, there are significant problems in the calculation, presentation and interpretation of the results that will need to be addressed. In general, some claims are not clearly demonstrated in the corresponding figures, conclusions are often contradictory throughout the paper and some figures have interesting patterns that are not clearly addressed. Also, the discussion about uncertainty is not integrated in the interpretation of the results, leading to constant contradictory comments.

**General Comments**

For equation 2: if for any given month $n$, the monthly change in water storage is defined as

$$dS_n = TWSA_n - TWSA_{n-1}$$

then, the three-months sliding window should be

$$dS_n = TWSA_{n+1} - TWSA_{n-2}$$

assuming that the window includes both month that precedes and follows month n. This should be corrected and ET recalculated accordingly.

Throughout the paper, it would be useful to have radiation, ET and rainfall in the same units, to be able to see the relative magnitudes of those values. This is especially true when doing a Radiation/Rainfall ratio, as in figure 5, or a scatterplot as in figure 6, and would greatly improve the legibility in figure 3 as well.

Every term in the water balance, as well as the additional terms of radiation, PET and EVI come from different datasets, with different assumptions, instruments and uncertainties. Even if it is very difficult to assess the overall uncertainty of the ET calculation, data could show problems that are only mentioned at the end of the paper, but never integrated in the result discussion, particularly when it is hard to make physical sense of the results. As examples, there are negative values in figure 4 (Solimões) or an unexplained decrease in ET in Solimões for August-November.

Even through the authors claim to find some water-limitation in a few basins, those claims are not well supported by their results, particularly in figure 3. Even though ET > P in Purus, Madeira and Tapajós for the months of May-September, ET is generally increasing in those months, showing that plants have access to water storage and are probably not water stressed, but respond to an increase in radiation. This is also shown in figure 4, where ET < P-dS for all basins and all months. There should be further discussion about PET > ET (figure 4) for Purus, Madeira and Tapajós for the months of May-September. The addition of MODIS PET in figure 3 could help with that discussion.

The paper makes a couple of claims about inter-annual variability of climatic factors and ET (e.g. page 11, line 6) without clearly analyzing or showing the relative magnitudes of those variabilities. The paper can either clearly analyze the inter-annual variability and discuss it, or drop the claims.

**Specific comments**

Figure 3 would highly benefit of having consistent units between radiation, precipitation and evapotranspiration. It would also benefit of a superposition of radiation and ET. The addition of runoff would add important information. Even though runoff is a major part of the model equation, it is never shown in the figures, making it harder to understand the entire water balance. Also, the figure should probably show dS/dt and not TWSA, which is the term used in the equation and the next figure.

Another interesting addition to figure 3 would be to add MODIS PET. It is used in figure 4 to calculate the Budyko dryness index, but understanding its relationship to ET in figure 3 would be very useful. A graph with radiation (unit converted), ET and PET together would be beneficial.

Figure 4 has inconsistencies. There are problematic negative values of AET/P-dS in the Solimões graph that are not present in the all-basin graph. If those values are correct, there must be problems in the data that are not mentioned.

Figure 4 is the main source indicating the water-limitation in Purus, Madeira and Tapajós. Even though, PET/P-dS > 1, most corresponding values of ET/P-dS are not close to 1, suggesting that there is a lot of water that is not used/available for ET, or that ET is underestimated. More discussion should address this point. The addition of another source of PET (derived from the Radiation data, with a Priesley-Taylor equation for example) could also add some information.

Figure 5 is very misleading. First the units should be corrected so that the ratio has some physical consistency and would represent some fashion of a dryness index. Even though the seasonality should not be affected by the unit conversion, the interpretation of this figure is weak. The decline in the radiation/rainfall ratio from August to October is due to a higher increase in rainfall than radiation but both values are increasing according to figure 3. If ET would "maximize its use of radiation and water", as stated on page 7, lines 21-22, then ET would still increase during those months, having both more radiation and more water available. Figure 3 shows that ET is always smaller than P, which suggests, as the authors pointed out (page 10, line 25), that the basin is never water-limited. Figure 3 also shows that ET is slightly out of phase with Radiation, which is puzzling and should be discussed. In short, figure 5 does not add valuable physical understanding of ET in Solimões, and more discussion should address the intriguing results of that particular basin (especially in the light of figure 2, low ET).

Figure 6 aggregates all basins into one figure, which can be debatable if there are to be differences in ET responses to climatic drivers in different regions. Nonetheless, the lowest ET values are mainly located in the mid-range of both radiation and rainfall, which is not discussed. High ET in relative low radiation and high precipitation could indicate that the ET is not limited by either climatic factor. On the other hand, the relatively low ET values when both radiation and precipitation are low can indicate a combined effect or either one of them, but with the units not being consistent it is hard to interpret. In short, the figure is not discussed in its full potential, and claims are weak.

Page 6, line 25-26. The sentence "Furthermore, …" is unclear because if part of the basin is at higher elevation with less rainfall, then it does not explain why the annual rainfall is still similar to that in Purus, or the relatively low ET.

Page 7, lines 1-3. ET>rainfall does not indicate a limitation in water availability, because of possible water storage and root access to deep water. Even if the lowest ET values were found when rainfall was in decline, ET increases in the lowest rainfall months. Those statements need to be rephrased.

Page 7, lines 8-10. There is confusion between seasonal and inter-annual variation in that first sentence. ET shows small seasonal variation (or none) but huge inter-annual variation (figure 3), but it does not read that way! As stated, that basin is probably operating at PET all year, having enough energy and water all the time (but that could be interesting to see, with added PET on figure 3)

Page 7, lines 19-22. See comments for Figure 5 above. If "ET maximizes the use of both solar radiation and water" then why is ET decreasing when both Radiation and rainfall increase? More discussion or analysis are needed to understand this region.

Page 7, lines 23-27. See comments for Figure 6 above.

Paragraph 3.3 is very interesting. More emphasis could be put on the fact that the relationship between greening and ET is weak (low $r^2$ and low/no slopes). The best correlation is in Solimões where the relation is opposite to the one in the other basins. In Tapajós, there is no correlation between EVI and ET, if anything ET is more variable (scattered) with higher EVI.

Page 8, lines 15-18. In the Solimões basin, the two models agree in their seasonal pattern of ET. They can be both misrepresenting what is happening, but given that the water-balance estimate of ET is hard to understand, some discussion about the water-balance method uncertainties (here or in paragraph 4.3) seems necessary.

Page 11, lines 10-15. Even though water balance methods are adequate alternative, the entire discussion lacks any perspective on uncertainties. In particular in places were ET seasonality is difficult to understand, like in Solimões.

Page 9, lines 26-30. This discussion would suggest that there is no water-limitation. This is inconsistent with figure 4 and some previous comments and discussion. It is also directly contradicted in page 10, line 9.

Page 10, line 25-26 is inconsistent with the statement on page 10, line 9, but in agreement with the previous discussion on page 9.

Paragraph 4.2. The discussion is interesting and maybe it could be more clearly stated that ET and EVI don't seem to have any causal relationship. Therefore using EVI for modeling ET is not advisable.

Page 11, line 5-6. There is no inter-annual analysis in this paper.

Page 11, line 7. See comment for figure 5. This statement is not backed by data.

Page 12, lines 3-13. This discussion should be integrated in the interpretation of figure 4, as suggested above. PET is likely overestimated in Madeira, Purus and Tapajós as well as Solimões, as discussed on page 9, lines 26-30.

**Minor comments**

Equation 1, page 3 line 27, should read

$$ET = P - R - \frac{dS}{dt}$$

time is in a lowercase format by convention (needs to be corrected in the following line 28 as well)

Page 9, lines 16-17 should read P-dS (capital S)

If possible, clean the connection of the upper/lower parts on figure 1 (around 13S) to eliminate the artificial line/disconnect.

Figure 4

- A great proportion of the figure is blank. It would be much clearer if the scale on the y-axis was only up to 1, therefore spreading out the data points
- Why is the graph not positioned at (0,0)? Why are there negative values of AET?
- The legend doesn't need to be repeated 5 times, thus gaining some space for the data

Figure 7. The values of Table 2 could be added to the figure, then the reader would not need to go back and forth from both of them.

Page 12, line 3 "therefore" means thus and should be spelled "therefore".

---

## Author Comment (AC1)

We would like to thank Dr Matthew Rodell for his excellent suggestions and comments. We are confident that, after addressing the reviewer's comments, the manuscript will be greatly improved. For instance, we will expand and improve the description of our methods, to avoid any misinterpretation on how the components of the water balance were obtained. Furthermore, as suggested by Dr Rodell, we will include uncertainty analysis for the ET estimates, which can be presented in several figures (including extra figures in the supplementary material) and discussed in the text.

Detailed replies for each comment are provided below:

**Comment**: Equation 2 is incorrect. It should be

$dSn = (TWSAn+1 − TWSAn−1)/((n+1)-(n-1))$, or equivalently,
$dSn = (TWSAn+1 − TWSAn−1)/2$,
as shown in equation 13 in Swenson and Wahr (2006). This error leads to dS, and consequently monthly/seasonal ET, being inaccurately estimated throughout the manuscript.

**Reply**: We believe a lack of details in the description of our methods might have misled the reviewer. We will improve this section to clarify the dS/dt calculations. Equation 2 refers only to the *dS* component, which is later divided by *dt*. Hence, the *dS/dT* component was indeed calculated similarly as in Swenson and Wahr (2006), in the sense that we have properly accounted for the inherent temporal sampling of GRACE. In our case, as the unit used in our water balance equation was mm month$^{-1}$, we have first divided *dS* by the number of days between GRACE observations, and then multiplied by the number of days in the month. Another detail is that we performed centered differences for calculating *dS* (instead of forward or backward differences, as suggested in Landerer et al 2010). For this, we have adjusted the TWSA values for the beginning ($TWSA_{n−1}$) and end ($TWSA_{n+1}$) of the respective months, resulting in a *dt* of 3 months, consistent with the three-month sliding window used for P and R. As pointed out by the reviewer, failing to correctly calculate the changes in water storage would lead to inaccurate ET estimates, which would probably be evident in our results.

**Comment**: There is no uncertainty analysis provided for the water budget ET estimates. It should be included, and could be accomplished by computing the square root of the sum of the squares of the P, R, and dS errors. See, for example, Rodell et al. (2011).

**Reply**: This was an excellent suggestion. We will carry out the uncertainty analysis as suggested. The calculations will be made as described in Rodell et al. (2011), but with a small difference in the assessments of rainfall errors. In our case, we will assess rainfall errors using TRMM 3B43 relative errors layer. Interestingly, after making preliminary tests, we observe that rainfall is shown to be the main source of uncertainty. We will then add the 95% confidence limits in the figures showing the ET seasonal pattern. The uncertainties for each component of the water balance will be shown in new figures at the supplementary material.

**Comment**: Page 2, line 30 – Add to the literature review that Rodell et al. (2011) applied the water balance approach (with observed precipitation, runoff, and GRACE terrestrial water storage) to ET estimation over the Tocantins River basin (among others) and found that the seasonal cycle of ET in that basin is weak.

**Reply**: We will add this literature review. Nonetheless, we would like to point out the fact that the Tocantins River basin is not part of the Amazon River basin. The vegetation cover across the Tocantins River basin is also quite different, being mostly covered by Cerrado (Savannas).

**Comment:** Page 7, line 6 – "are" should be "is" (the subject is "analysis").

**Reply**: Will be corrected as suggested.

---

## Author Comment (AC2)

We would like to thank the reviewer for providing valuable comments and suggestions to improve our manuscript. The points raised by the reviewer were very relevant. We are confident we can address all comments and we believe the manuscript will be largely improved after the corrections.

Detailed replies for each comment are provided below.

**Comment**:

For equation 2: if for any given month n, the monthly change in water storage is defined as

$dS_n = TWSA_n - TWSA_{n-1}$

then, the three-months sliding window should be

$dS_n = TWSA_{n+1} - TWSA_{n-2}$

assuming that the window includes both month that precedes and follows month n. This should be corrected and ET recalculated accordingly.

**Reply**:

This is probably a misunderstanding. We will improve the description of the methods to clarify this point. Given that GRACE provides monthly mean TWS anomalies, the dS calculation has to be adjusted to account for this temporal sampling. There are different approaches for calculating dS, including forward and backward differences (Landerer et al 2010), as suggested by the reviewer. However, due to errors and uncertainties in GRACE data, these approaches may result in high-frequency artifacts (Landerer et al 2010). To overcome this problem, we carried out centered differences for dS calculation. That is, dS for a given month *n* was estimated considering the accumulated fluxes from the beginning of month *n-1* to the end of month *n+1*. Hence, we considered changes in TWSA from the first day of month n-1 to the last day of month n+1 (i.e. separated by 3 months). For this procedure, TWSA values had to be adjusted for the beginning and end of each month, as described in page 4.

Landerer, F. W., J. O. Dickey, and A. Güntner (2010), Terrestrial water budget of the Eurasian pan-Arctic from GRACE satellite measurements during 2003–2009, J. Geophys. Res., 115, D23115, doi:10.1029/2010JD014584.

**Comment**: Figure 3 would highly benefit of having consistent units between radiation, precipitation and evapotranspiration. It would also benefit of a superposition of radiation and ET. The addition of runoff would add important information. Even though runoff is a major part of the model equation, it is never shown in the figures, making it harder to understand the entire water balance. Also, the figure should probably show dS/dt and not TWSA, which is the term used in the equation and the next figure.

**Reply**: These are good suggestions. We will change the radiation unit to equivalent evaporation i.e. mm month[-1]. Furthermore, we will add new figures in the supplementary material, showing all the components of the water balance, including runoff and dS/dt, as you suggested. This information will not be added in Figure 3, to avoid overloading the plots. For instance, superposing radiation and ET would hinder the visualization of ET patterns, which are the most important information in this figure. We will, however, include a figure superposing radiation and ET in the supplementary material (Fig. S2).

**Comment**: Another interesting addition to figure 3 would be to add MODIS PET. It is used in figure 4 to calculate the Budyko dryness index, but understanding its relationship to ET in figure 3 would be very useful. A graph with radiation (unit converted), ET and PET together would be beneficial.

**Reply**: The suggested plots will be added in the supplementary material.

**Comment**: Figure 4 has inconsistencies. There are problematic negative values of AET/P-dS in the Solimões graph that are not present in the all-basin graph. If those values are correct, there must be problems in the data that are not mentioned.

**Reply**: This is perhaps a misunderstanding. There aren't any negative values of AET/P-dS in the plots. This confusion was probably caused because a slight difference in the y axis origin. The origin was set to zero for the "all-basin" plots, while in the plots for the individual basins, the origin was set to 0.2. The plots will be corrected, and the figure will be improved based on your comments (below), which will certainly contribute to solve this confusion.

Figure 4

- A great proportion of the figure is blank. It would be much clearer if the scale on the y-axis was only up to 1, therefore spreading out the data points

- Why is the graph not positioned at (0,0)? Why are there negative values of AET?

- The legend doesn't need to be repeated 5 times, thus gaining some space for the data

**Reply**: The figure will be modified as suggested. Y-axis limit will be set to 1.2, and the origin will be set at (0,0). The repeated legends will be removed from the plot, and a single legend will be inserted under the figure.

**Comment**: Figure 4 is the main source indicating the water-limitation in Purus, Madeira and Tapajós. Even though, PET/P-dS > 1, most corresponding values of ET/P-dS are not close to 1, suggesting that there is a lot of water that is not used/available for ET, or that ET is underestimated. More discussion should address this point. The addition of another source of PET (derived from the Radiation data, with a Priesley-Taylor equation for example) could also add some information.

**Reply**. An additional Budyko plot, made using an alternative source of PET, calculated from a variant of the Penman–Monteith formula, will be included in the supplementary material (Fig S3). This new PET dataset will be obtained from the high-resolution grids of monthly climatic observations (CRU TS3.23) (Harris et al., 2014). The grid is constructed using monthly observations at meteorological stations. Due to the sparse network of ground meteorological stations over the amazon region, we will maintain the MODIS PET data as the primary dataset in the manuscript.

Harris, I., Jones, P. D., Osborn, T. J., & Lister, D. H. (2014). Updated high-resolution grids of monthly climatic observations - the CRU TS3.10 Dataset. International Journal of Climatology, 34(3), 623–642. https://doi.org/10.1002/joc.3711

**Comment**: In short, figure 5 does not add valuable physical understanding of ET in Solimões...

**Reply**: Figure 5 will be removed from the manuscript.

**Comment**: Figure 6 aggregates all basins into one figure, which can be debatable if there are to be differences in ET responses to climatic drivers in different regions. Nonetheless, the lowest ET values are mainly located in the mid-range of both radiation and rainfall, which is not discussed. High ET in relative low radiation and high precipitation could indicate that the ET is not limited by either climatic factor. On the other hand, the relatively low ET values when both radiation and precipitation are low can indicate a combined effect or either one of them, but with the units not being consistent it is hard to interpret. In short, the figure is not discussed in its full potential, and claims are weak.

Page 7, lines 23-27. See comments for Figure 6 above.

**Reply**: These are good suggestions, we agree with the reviewer. We will change the radiation units in Figure 6 to same unit as rainfall to facilitate comparison. We will also improve the description and discussion of these results, as suggested by the reviewer. We consider that the fact that the lowest ET values are located in the mid-range of both radiation and rainfall are likely due to the influence of other variables, in particular soil water storage and root access to deep water. This argument is reinforced by the fact that at this radiation range (~ 200 mm month$^{-1}$), when rainfall drops below 100 mm month$^{-1}$, an increase in ET cannot be explained by water input from rainfall. The discussion of the figure will be further developed to clarify this argument.

**Comment**: Page 6, line 25-26. The sentence "Furthermore, …" is unclear because if part of the basin is at higher elevation with less rainfall, then it does not explain why the annual rainfall is still similar to that in Purus, or the relatively low ET.

**Reply**: In fact, the mean annual rainfall rates cannot provide enough information to explain these differences in ET. As shown in our Figure 1, the spatial distribution of rainfall rates in these two basins varies significantly. While rainfall in Purus is uniformly distributed (~2000 mm year$^{-1}$), in the Solimões basin we see a mixture of very wet (>3500 mm year$^{-1}$) and dry (<500 mm year$^{-1}$). The sentence you are referring to, is simply pointing to this spatial heterogeneity as a possible cause for the differences in ET.

**Comment**: Page 7, lines 1-3. ET>rainfall does not indicate a limitation in water availability, because of possible water storage and root access to deep water. Even if the lowest ET values were found when rainfall was in decline, ET increases in the lowest rainfall months. Those statements need to be rephrased.

**Reply**: We agree with the reviewer. These statements will be rephrased to account for the possible influence of soil water storage and root access to deep water in compensating the rainfall deficit.

**Comment**: Page 7, lines 8-10. There is confusion between seasonal and inter-annual variation in that first sentence. ET shows small seasonal variation (or none) but huge inter-annual variation (figure 3), but it does not read that way! As stated, that basin is probably operating at PET all year,

having enough energy and water all the time (but that could be interesting to see, with added PET on figure 3).

**Reply**: This sentence does not contain any remark about inter-annual variation. The sentence mentioned by the reviewer refers to "intra-annual" variation, that is, variations occurring on a time scale of less than 1 year.

**Comment**: Paragraph 3.3 is very interesting. More emphasis could be put on the fact that the relationship between greening and ET is weak (low r2 and low/no slopes). The best correlation is in Solimões where the relation is opposite to the one in the other basins. In Tapajós, there is no correlation between EVI and ET, if anything ET is more variable (scattered) with higher EVI.

**Reply**: We agree that these are interesting results. We will further develop the manuscript to include improved discussion on this specific topic. In particular, we will emphasize the biophysical causes of EVI seasonality, and how they may be related to ET. We will also exam previous studies evaluating the relationship between forest functioning and EVI, to discuss how our results may contribute to better understand climate-vegetation interactions in this region.

**Comment**: Page 8, lines 15-18. In the Solimões basin, the two models agree in their seasonal pattern of ET. They can be both misrepresenting what is happening, but given that the water-balance estimate of ET is hard to understand, some discussion about the water-balance method uncertainties (here or in paragraph 4.3) seems necessary.

Page 11, lines 10-15. Even though water balance methods are adequate alternative, the entire discussion lacks any perspective on uncertainties. In particular in places were ET seasonality is difficult to understand, like in Solimões.

**Reply**: We agree. We plan to extensively improve the assessment of uncertainties in the manuscript. We will include the 95% confidence intervals of ET estimates in figures 3 and 9. The uncertainties will be calculated by combining measurement errors on P, R, and dS/dt, as suggested in Rodell et al (2011). The approach will be fully described in the methods section. Errors in GRACE TWSA estimates will be assessed using gridded fields of measurement and leakage errors provided with GRC Tellus data (Landerer and Swenson, 2012), while uncertainties in monthly rainfall values will be assessed using the rainfall relative error layer available in the TRMM 3B43 product (Huffman, 1997).

*Huffman, G. J.: Estimates of Root-Mean-Square Random Error for Finite Samples of Estimated Precipitation, J. Appl. Meteorol., 36, 1191–1201, doi:10.1175/1520-0450(1997)036<1191:EORMSR>2.0.CO;2, 1997.*

*Landerer, F. W. and Swenson, S. C.: Accuracy of scaled GRACE terrestrial water storage estimates, Water Resour. Res., 48(4), 1–11, doi:10.1029/2011WR011453, 2012.*

**Comment**: Page 9, lines 26-30. This discussion would suggest that there is no water-limitation. This is inconsistent with figure 4 and some previous comments and discussion. It is also directly contradicted in page 10, line 9.

Page 10, line 25-26 is inconsistent with the statement on page 10, line 9, but in agreement with the previous discussion on page 9.

**Reply**: The reviewer is correct, page 10, line 9 contained conflicting information. We will correct this mistake, so the sentence will be revised as: *"…the Madeira and Tapajós basins are characterized by a large latitudinal gradient and, consequently, different ecosystems are present within these sub-basins. Hence, it is likely that, although on average water availability is not critical at the Tapajós and Madeira basins during the dry season, water limitation may occur in southern (drier) parts of these basins."*

We highlight that this paragraph is discussing spatial variabilities within basins, and not between basins. That is, because some of these basins have large latitudinal gradient, the southern (drier) portion of these basin may suffer from water limitation, while the northern (wetter) portions may not. This spatial variability is likely masked when assessing averaged ET for the entire basin.

**Comment**: Paragraph 4.2. The discussion is interesting and maybe it could be more clearly stated that ET and EVI don't seem to have any causal relationship. Therefore using EVI for modeling ET is not advisable.

**Reply**: We agree. We will include one more paragraph at section 4.2, to discuss the implications of the relationship between ET and EVI for modelling applications.

**Comment**: Page 11, line 5-6. There is no inter-annual analysis in this paper.

**Reply**: This is again a misunderstanding. The sentence is referring to "intra-annual variability" and not inter-annual.

**Comment**: Page 11, line 7. See comment for figure 5. This statement is not backed by data.

**Reply**: We will remove this statement.

**Comment**: Page 12, lines 3-13. This discussion should be integrated in the interpretation of figure 4, as suggested above. PET is likely overestimated in Madeira, Purus and Tapajós as well as Solimões, as discussed on page 9, lines 26-30.

**Reply**: Although we appreciate the reviewer's suggestion, this is a debatable argument, as it is dependent on writing style. One may argue that any type of discussion should be kept separated from the results. For instance, guidelines from Columbia University clearly advises that the Results chapter should "not discuss the results or speculate as to why something happened". Similar suggestion is repeated by guidelines published in an editorial in Nature Structural & Molecular Biology, and other sources.

http://www.nature.com/nsmb/journal/v17/n2/full/nsmb0210-139.html

http://www.columbia.edu/cu/biology/ug/research/paper.html

**Minor comments**

Equation 1, page 3 line 27, time is in a lowercase format by convention (needs to be corrected in the following line 28 as well)

Reply: Will be corrected.

Page 9, lines 16-17 should read P-dS (capital S)

Reply: Will be corrected.

If possible, clean the connection of the upper/lower parts on figure 1 (around 13S) to eliminate the artificial line/disconnect.

Reply: This is an artifact created during file conversion. The original figure doesn't have this line.

Figure 7. The values of Table 2 could be added to the figure, then the reader would not need to go back and forth from both of them.

Reply: In fact, the first version of this figure was exactly as requested, but after some tests, all authors agreed that the information is more clearly presented when the coefficients and $R^2$ are presented separately in a table.

Page 12, line 3 "therefore" means thus and should be spelled "therefore".

Reply: Will be fixed.

---

## Author Response (AR1)

We would like to thank Dr Matthew Rodell and one anonymous reviewer for their excellent suggestions and comments. We are confident that the manuscript has been greatly improved after addressing their suggestions. We have expanded and improved the description of our methods, to avoid any misinterpretation on how the components of the water balance were obtained. Furthermore, as suggested by Dr Rodell, we included uncertainty analysis for the ET estimates, which is now presented in several figures (including extra figures in the supplementary material) and discussed in the text.

Detailed replies for each comment are provided below:

Comment: Equation 2 is incorrect. It should be

dSn = (TWSAn+1 - TWSAn-1)/((n+1)-(n-1)), or equivalently, dSn = (TWSAn+1 - TWSAn-1)/2, as shown in equation 13 in Swenson and Wahr (2006). This error leads to dS, and consequently monthly/seasonal ET, being inaccurately estimated throughout the manuscript.

**Reply**: We believe a lack of details in the description of our methods might have misled the reviewer. We have improved this section to clarify the dS/dt calculations. Equation 2 refers only to the *dS* component, which is later divided by *dt*. Hence, the *dS*/*dT* component was indeed calculated similarly as in Swenson and Wahr (2006), in the sense that we have properly accounted for the inherent temporal sampling of GRACE. In our case, as the unit used in our water balance equation was mm month-1, we have first divided *dS* by the number of days between GRACE observations, and then multiplied by the number of days in the month. Another detail is that we performed centered differences for calculating *dS* (instead of forward or backward differences, as suggested in Landerer et al 2010). For this, we have adjusted the TWSA values for the beginning (*TWSA**n*-1) and end (*TWSA**n*+1) of the respective months, resulting in a *dt* of 3 months, consistent with the three-month sliding window used for P and R. As pointed out by the reviewer, failing to correctly calculate the changes in water storage would lead to inaccurate ET estimates, which would probably be evident in our results.

**Comment**: There is no uncertainty analysis provided for the water budget ET estimates. It should be included, and could be accomplished by computing the square root of the sum of the squares of the P, R, and dS errors. See, for example, Rodell et al. (2011).

**Reply**: This was an excellent suggestion. We carried out the uncertainty analysis as suggested. The calculations were made as described in Rodell et al. (2011), but with a small difference in the assessments of rainfall errors. In our case, we assessed rainfall errors using TRMM 3B43 relative errors layer. Interestingly, we observe that rainfall is shown to be the main source of uncertainty. We have added the 95% confidence limits in the figures showing the ET seasonal pattern. The uncertainties for each component of the water balance are shown in new figures at the supplementary material.

**Comment**: Page 2, line 30 - Add to the literature review that Rodell et al. (2011) applied the water balance approach (with observed precipitation, runoff, and GRACE terrestrial water storage) to ET estimation over the Tocantins River basin (among others) and found that the seasonal cycle of ET in that basin is weak.

**Reply**: We have added this literature review. Nonetheless, we would like to point out the fact that the Tocantins River basin is not part of the Amazon River basin. The vegetation cover across the Tocantins River basin is also quite different, being mostly covered by Cerrado (Savannas).

Comment: Page 7, line 6 – "are" should be "is" (the subject is "analysis").

Reply: Corrected as suggested.

**Reviewer #2**

**Comment:**

For equation 2: if for any given month n, the monthly change in water storage is defined as

dSn=TWSAn-TWSAn-1

then, the three-months sliding window should be

dSn=TWSAn+1-TWSAn-2

assuming that the window includes both month that precedes and follows month n. This should be corrected and ET recalculated accordingly.

**Reply:**

This is probably a misunderstanding. We improved the description of the methods to clarify this point. Given that GRACE provides monthly mean TWS anomalies, the dS calculation has to be adjusted to account for this temporal sampling. There are different approaches for calculating dS, including forward and backward differences (Landerer et al 2010), as suggested by the reviewer. However, due to errors and uncertainties in GRACE data, these approaches may result in high-frequency artifacts (Landerer et al 2010). To overcome this problem, we carried out centered differences for dS calculation. That is, dS for a given month *n* was estimated considering the accumulated fluxes from the beginning of month *n*-1 to the end of month *n*+1. Hence, we considered changes in TWSA from the first day of month n-1 to the last day of month n+1 (i.e. separated by 3 months). For this procedure, TWSA values had to be adjusted for the beginning and end of each month, as described in page 4.

Landerer, F. W., J. O. Dickey, and A. Güntner (2010), Terrestrial water budget of the Eurasian pan-Arctic from GRACE satellite measurements during 2003–2009, J. Geophys. Res., 115, D23115, doi:10.1029/2010JD014584.

**Comment**: Figure 3 would highly benefit of having consistent units between radiation, precipitation and evapotranspiration. It would also benefit of a superposition of radiation and ET. The addition of runoff would add important information. Even though runoff is a major part of the model equation, it is never shown in the figures, making it harder to understand the entire water balance. Also, the figure should probably show dS/dt and not TWSA, which is the term used in the equation and the next figure.

**Reply**: These are good suggestions. We changed the radiation unit to equivalent evaporation i.e. mm month-1. Furthermore, we added new figures in the supplementary material, showing all the components of the water balance, including runoff and dS/dt, as you suggested. This information was not added in Figure 3, to avoid overloading the plots. For instance, superposing radiation and ET would hinder the visualization of ET patterns, which are the most important information in this figure. We, however, included a figure superposing radiation and ET in the supplementary material (Fig. S2).

**Comment**: Another interesting addition to figure 3 would be to add MODIS PET. It is used in figure 4 to calculate the Budyko dryness index, but understanding its relationship to ET in figure 3 would be very useful. A graph with radiation (unit converted), ET and PET together would be beneficial.

**Reply**: The suggested plots were added in the supplementary material.

**Comment**: Figure 4 has inconsistencies. There are problematic negative values of AET/P-dS in the Solimões graph that are not present in the all-basin graph. If those values are correct, there must be problems in the data that are not mentioned.

**Reply**: This is perhaps a misunderstanding. There aren't any negative values of AET/P-dS in the plots. This confusion was probably caused because a slight difference in the y axis origin. The origin was set to zero for the "all-basin" plots, while in the plots for the individual basins, the origin was set to 0.2. The plots were corrected, and the figure was improved based on your comments (below).

**Figure 4**

- A great proportion of the figure is blank. It would be much clearer if the scale on the y-axis was only up to 1, therefore spreading out the data points

- Why is the graph not positioned at (0,0)? Why are there negative values of AET?

- The legend doesn't need to be repeated 5 times, thus gaining some space for the data

**Reply**: The figure was modified as suggested. Y-axis limit was set to 1.2, and the origin was set at (0,0). The repeated legends were removed from the plot, and a single legend inserted under the figure.

**Comment**: Figure 4 is the main source indicating the water-limitation in Purus, Madeira and Tapajós. Even though, PET/P-dS > 1, most corresponding values of ET/P-dS are not close to 1, suggesting that there is a lot of water that is not used/available for ET, or that ET is underestimated. More discussion should address this point. The addition of another source of PET (derived from the Radiation data, with a Priesley-Taylor equation for example) could also add some information.

**Reply**. An additional Budyko plot, made using an alternative source of PET, calculated from a variant of the Penman–Monteith formula, was included in the supplementary material (Fig S3). This new PET dataset was obtained from the high-resolution grids of monthly climatic observations (CRU TS3.23) (Harris et al., 2014). The grid is constructed using monthly observations at meteorological stations. Due to the sparse network of ground meteorological stations over the amazon region, we will maintain the MODIS PET data as the primary dataset in the manuscript.

Harris, I., Jones, P. D., Osborn, T. J., & Lister, D. H. (2014). Updated high-resolution grids of monthly climatic observations - the CRU TS3.10 Dataset. International Journal of Climatology, 34(3), 623–642. https://doi.org/10.1002/joc.3711

**Comment**: In short, figure 5 does not add valuable physical understanding of ET in Solimões...

**Reply**: Figure 5 was removed from the manuscript.

**Comment**: Figure 6 aggregates all basins into one figure, which can be debatable if there are to be differences in ET responses to climatic drivers in different regions. Nonetheless, the lowest ET values are mainly located in the mid-range of both radiation and rainfall, which is not discussed. High ET in relative low radiation and high precipitation could indicate that the ET is not limited by either climatic factor. On the other hand, the relatively low ET values when both radiation and precipitation are low can indicate a combined effect or either one of them, but with the units not being consistent it is hard to interpret. In short, the figure is not discussed in its full potential, and claims are weak.

Page 7, lines 23-27. See comments for Figure 6 above.

**Reply**: These are good suggestions, we agree with the reviewer. We changed the radiation units in Figure 6 to same unit as rainfall to facilitate comparison. We also improved the description and discussion of these results, as suggested by the reviewer. We consider that the fact that the lowest ET values are located in the mid-range of both radiation and rainfall are likely due to the influence of other variables, in particular soil water storage and root access to deep water. This argument is reinforced by the fact that at this radiation range (~ 200 mm month-1), when rainfall drops below 100 mm month-1, an increase in ET cannot be explained by water input from rainfall. The discussion of the figure has been further developed to clarify this argument.

**Comment**: Page 6, line 25-26. The sentence "Furthermore, ..." is unclear because if part of the basin is at higher elevation with less rainfall, then it does not explain why the annual rainfall is still similar to that in Purus, or the relatively low ET.

**Reply**: In fact, the mean annual rainfall rates cannot provide enough information to explain these differences in ET. As shown in our Figure 1, the spatial distribution of rainfall rates in these two basins varies significantly. While rainfall in Purus is uniformly distributed (~2000 mm year-1), in the Solimões basin we see a mixture of very wet (>3500 mm year-1) and dry (<500 mm year-1). The sentence you are referring to, is simply pointing to this spatial heterogeneity as a possible cause for the differences in ET.

**Comment**: Page 7, lines 1-3. ET>rainfall does not indicate a limitation in water availability, because of possible water storage and root access to deep water. Even if the lowest ET values were found when rainfall was in decline, ET increases in the lowest rainfall months. Those statements need to be rephrased.

**Reply**: We agree with the reviewer. These statements were rephrased to account for the possible influence of soil water storage and root access to deep water in compensating the rainfall deficit.

**Comment**: Page 7, lines 8-10. There is confusion between seasonal and inter-annual variation in that first sentence. ET shows small seasonal variation (or none) but huge inter-annual variation (figure 3), but it does not read that way! As stated, that basin is probably operating at PET all year, having enough energy and water all the time (but that could be interesting to see, with added PET on figure 3).

**Reply**: This sentence does not contain any remark about inter-annual variation. The sentence mentioned by the reviewer refers to "intra-annual" variation, that is, variations occurring on a time scale of less than 1 year.

**Comment**: Paragraph 3.3 is very interesting. More emphasis could be put on the fact that the relationship between greening and ET is weak (low r2 and low/no slopes). The best correlation is in Solimões where the relation is opposite to the one in the other basins. In Tapajós, there is no correlation between EVI and ET, if anything ET is more variable (scattered) with higher EVI.

**Reply**: We agree that these are interesting results. We have further developed the manuscript to include improved discussion on this specific topic. In particular, we have emphasized the biophysical causes of EVI seasonality, and how they may be related to ET. We also examined previous studies evaluating the relationship between forest functioning and EVI, to discuss how our results may contribute to better understand climate-vegetation interactions in this region.

**Comment**: Page 8, lines 15-18. In the Solimões basin, the two models agree in their seasonal pattern of ET. They can be both misrepresenting what is happening, but given that the water-balance estimate of ET is hard to understand, some discussion about the water-balance method uncertainties (here or in paragraph 4.3) seems necessary.

Page 11, lines 10-15. Even though water balance methods are adequate alternative, the entire discussion lacks any perspective on uncertainties. In particular in places were ET seasonality is difficult to understand, like in Solimões.

**Reply**: We agree. We have extensively improved the assessment of uncertainties in the manuscript. We included the 95% confidence intervals of ET estimates in figures 3 and 9. The uncertainties were calculated by combining measurement errors on P, R, and dS/dt, as suggested in Rodell et al (2011). The approach has been fully described in the methods section. Errors in GRACE TWSA estimates were assessed using gridded fields of measurement and leakage errors provided with GRC Tellus data (Landerer and Swenson, 2012), while uncertainties in monthly rainfall values were assessed using the rainfall relative error layer available in the TRMM 3B43 product (Huffman, 1997).

Huffman, G. J.: Estimates of Root-Mean-Square Random Error for Finite Samples of Estimated Precipitation, J. Appl. Meteorol., 36, 1191–1201, doi:10.1175/1520-0450(1997)036<1191:EORMSR>2.0.CO;2, 1997.

Landerer, F. W. and Swenson, S. C.: Accuracy of scaled GRACE terrestrial water storage estimates, Water Resour. Res., 48(4), 1–11, doi:10.1029/2011WR011453, 2012.

**Comment**: Page 9, lines 26-30. This discussion would suggest that there is no water-limitation. This is inconsistent with figure 4 and some previous comments and discussion. It is also directly contradicted in page 10, line 9.

Page 10, line 25-26 is inconsistent with the statement on page 10, line 9, but in agreement with the previous discussion on page 9.

**Reply**: The reviewer is correct, page 10, line 9 contained conflicting information. We have corrected this mistake, so the sentence is revised as: *"…the Madeira and Tapajós basins are characterized by a large latitudinal gradient and, consequently, different ecosystems are present within these sub-basins. Hence, it is likely that, although on average water availability is not critical at the Tapajós and Madeira basins during the dry season, water limitation may occur in southern (drier) parts of these basins."*

We highlight that this paragraph is discussing spatial variabilities within basins, and not between basins. That is, because some of these basins have large latitudinal gradient, the southern (drier) portion of these basin may suffer from water limitation, while the northern (wetter) portions may not. This spatial variability is likely masked when assessing averaged ET for the entire basin.

**Comment**: Paragraph 4.2. The discussion is interesting and maybe it could be more clearly stated that ET and EVI don't seem to have any causal relationship. Therefore using EVI for modeling ET is not advisable.

**Reply**: We agree. We have included one more paragraph at section 4.2, to discuss the implications of the relationship between ET and EVI for modelling applications.

**Comment: Page 11, line 5-6. There is no inter-annual analysis in this paper.**

**Reply**: This is again a misunderstanding. The sentence is referring to "intra-annual variability" and not inter-annual.

**Comment**: Page 11, line 7. See comment for figure 5. This statement is not backed by data.

Reply: We removed this statement.

**Comment**: Page 12, lines 3-13. This discussion should be integrated in the interpretation of figure 4, as suggested above. PET is likely overestimated in Madeira, Purus and Tapajós as well as Solimões, as discussed on page 9, lines 26-30.

**Reply**: Although we appreciate the reviewer's suggestion, this is a debatable argument, as it is dependent on writing style. One may argue that any type of discussion should be kept separated from the results. For instance, guidelines from Columbia University clearly advises that the Results chapter should "not discuss the results or speculate as to why something happened". Similar

suggestion is repeated by guidelines published in an editorial in Nature Structural & Molecular Biology, and other sources.

http://www.nature.com/nsmb/journal/v17/n2/full/nsmb0210-139.html

http://www.columbia.edu/cu/biology/ug/research/paper.html

**Minor comments**

Equation 1, page 3 line 27, time is in a lowercase format by convention (needs to be corrected in the following line 28 as well)

**Reply: Corrected.**

Page 9, lines 16-17 should read P-dS (capital S)

**Reply: Corrected.**

If possible, clean the connection of the upper/lower parts on figure 1 (around 13S) to eliminate the artificial line/disconnect.

Reply: This is an artifact created during file conversion. The original figure doesn't have this line.

Figure 7. The values of Table 2 could be added to the figure, then the reader would not need to go back and forth from both of them.

Reply: In fact, the first version of this figure was exactly as requested, but after some tests, all authors agreed that the information is more clearly presented when the coefficients and R2 are presented separately in a table.

Page 12, line 3 "therefore" means thus and should be spelled "therefore".

Reply: Fixed.